# Improved estimation of cancer dependencies from large-scale RNAi screens using model-based normalization and data integration

James M. McFarland[1], Zandra V. Ho [1], Guillaume Kugener[1], Joshua M. Dempster[1], Phillip G. Montgomery[1], Jordan G. Bryan[1], John M. Krill-Burger [1], Thomas M. Green[1], Francisca Vazquez[1,2], Jesse S. Boehm [1], Todd R. Golub [1,2,3,4,5], William C. Hahn [1,2,3,6], David E. Root[1] & Aviad Tsherniak [1]

The availability of multiple datasets comprising genome-scale RNAi viability screens in hundreds of diverse cancer cell lines presents new opportunities for understanding cancer vulnerabilities. Integrated analyses of these data to assess differential dependency across genes and cell lines are challenging due to confounding factors such as batch effects and variable screen quality, as well as difficulty assessing gene dependency on an absolute scale. To address these issues, we incorporated cell line screen-quality parameters and hierarchical Bayesian inference into DEMETER2, an analytical framework for analyzing RNAi screens (https://depmap.org/R2-D2). This model substantially improves estimates of gene dependency across a range of performance measures, including identification of gold-standard essential genes and agreement with CRISPR/Cas9-based viability screens. It also allows us to integrate information across three large RNAi screening datasets, providing a unified resource representing the most extensive compilation of cancer cell line genetic dependencies to date.

[1] Broad Institute of MIT and Harvard, Cambridge 02142 MA, USA. [2] Dana-Farber Cancer Institute, Boston 02215 MA, USA. [3] Harvard Medical School, Boston 02115 MA, USA. [4] Boston Children's Hospital, Boston 02115 MA, USA. [5] Howard Hughes Medical Institute, Chevy Chase 20815 MD, USA. [6] Department of Medicine, Brigham and Women's Hospital, Boston 02115 MA, USA. Correspondence and requests for materials should be addressed to A.T. (email: aviad@broadinstitute.org)

Large-scale RNAi screens for cancer dependencies have recently been performed by multiple groups[1–3], providing systematic assessments of the effects of single-gene knockdown on cell viability, across a wide range of well-characterized cancer cell lines that are beginning to reflect the diversity of tumor types. By comparing genetic dependencies across cancer cell lines, researchers can thus identify specific cancer subtypes exhibiting a given vulnerability, as well as uncover new functional relationships between genes. In theory, integrating information across these separate RNAi datasets might greatly increase their utility—both by providing the broadest coverage of cell lines and genes assayed, as well as by improving the accuracy and precision of individual gene dependency estimates. However, such integration requires addressing several computational challenges.

Firstly, the presence of substantial off-target effects mediated by the microRNA pathway[4,5], as well as variable reagent efficacy, have long been recognized as challenges that can confound the interpretation of RNAi screening data. A number of methods have been developed to address these issues by utilizing robust statistics[6–8], mixed-effect models[3,9], or explicit models of microRNA-mediated effects[10,11]. Previously, we developed the DEMETER algorithm, a computational approach that models the "seed-sequence" specific off-target effect of each shRNA directly, along with variable shRNA efficacy[1]. While DEMETER and related approaches[8] provide improved isolation of on-target gene-knockdown effects, they assess only the relative differences in gene dependency across cell lines. This limitation precludes identification of genes that are "common essential" across cell lines, and makes direct comparisons of knockdown effects across genes difficult.

Another challenge with interpreting large-scale RNAi screens is that differences in screen quality between cell lines (as measured, for example, by the separation of positive and negative control gene dependencies) can confound comparisons of their genetic dependencies. Indeed, mRNA expression of *AGO2*, the catalytic component of the RNAi-inducing silencing complex (RISC), has been shown to correlate strongly with a cell line's screen quality[12], and is associated with the apparent essentiality of many genes[2]. This suggests that differences in the RNAi machinery between cell lines can bias quantification of the relative strength of their gene dependencies. Thus, it stands to reason that removal of such systematic screen-related differences might provide a more accurate estimation of the relevant patterns of genetic dependency.

Finally, the need to integrate RNAi screening datasets to generate a unified resource of cancer genetic dependencies raises several additional challenges. Such data integration requires analytical methods which can address batch effects at the cell line and shRNA level, and handle partial overlap of shRNAs and cell lines across datasets. Further, statistically principled models are needed in order to efficiently integrate evidence across datasets with variable screen quality, as well as differences in the number and quality of reagents targeting each gene.

We thus developed a method, DEMETER2 (D2) that builds on the DEMETER model to address these challenges. We demonstrate this approach by applying it to three of the largest published RNAi datasets, showing that it improves gene dependency estimates and allows for the effective integration of these data. The resulting combined dataset represents the most extensive compilation of cancer genetic dependencies to date, and will facilitate the discovery of therapeutic targets, as well as new cancer biology.

## Results

### Improved model of RNAi screening data. DEMETER2 is a model for large-scale pooled RNAi screening data, which takes as input measured changes in the relative abundance of pooled shRNA reagents across a panel of cell lines[1–3,7,13,14] and infers the effects of gene knockdown on the viability of each cell line. As in the original DEMETER (D1) model, DEMETER2 (D2) accounts for the depletion of each shRNA over time as a combination of the effects of suppressing the genes targeted by the shRNA, along with seed-based off-target effects determined by two 7-mer seed sequences within each shRNA[1]. The DEMETER models also estimate the efficacy of each shRNA in eliciting these gene- and seed-effects.

D2 builds on the original D1 model by adding several additional components (Fig. 1a), summarized here (see Methods for details). First, D2 estimates a "screen signal" parameter for each cell line, which accounts for overall differences in the relative strength of gene knockdown effects, such as due to variable RNAi efficacy[2,12,15,16]. The model also incorporates scaling and offset terms for each screen to account for global differences in the distributions of shRNA depletion levels (such as those produced by differences in the number of population doublings and passages between measurements). Furthermore, D2 estimates the noise level associated with each screen to account for variable data quality.

Another key addition in D2 is the use of a hierarchical model for the gene and seed effects, allowing for efficient pooling of information across cell lines. Combined with shRNA-specific terms designed to capture measurement errors in the initial shRNA abundance and unaccounted-for off-target effects, these additions allow the model to accurately estimate gene dependency on an absolute scale (where a zero score represents no dependency) rather than a relative scale as in D1 (where a zero score represents the average dependency across all cell lines).

Finally, D2 utilizes a Bayesian inference approach for parameter estimation which provides uncertainty estimates for the gene effects and other model parameters. In addition to facilitating comparisons of gene effects across cell lines and genes, where the precision of estimates can vary widely, these uncertainty estimates can be directly utilized in downstream analyses to improve their statistical power, as we demonstrate below.

### D2 accurately estimates absolute gene dependency. We first sought to compare D2 with existing methods in terms of its ability to identify genes which are essential in individual cell lines, as well as genes which are "common essential" across cellular contexts. To this end, we utilized two recently published RNAi datasets: the Broad Institute Project Achilles dataset[1], which consists of 501 cell lines screened with 94k shRNAs targeting 17k genes (with a median coverage of 5 shRNAs per gene), and the Novartis DRIVE dataset[2], which consists of 397 cell lines screened with 158k shRNAs targeting 8k genes (with a median coverage of 20 shRNAs per gene).

First, we measured the accuracy of dependency scores from each model by computing positive/negative control separation (measured by the strictly standardized mean difference, or SSMD), using a curated list of gold standard common-essential genes[17] as positive controls, and genes that were unexpressed in each cell line as negative controls. Since DEMETER[1] and ATARiS[8] only estimate relative differences in gene dependency across cell lines, they cannot be used for such analyses. Thus, we compare the performance of D2 with a simple approach that averages the depletion scores across shRNAs targeting each gene (gene averaging; GA).

Compared with GA, D2 provided much more accurate identification of essential genes in the Achilles dataset (SSMD increased by 58% on average, with improvement for all 486 cell

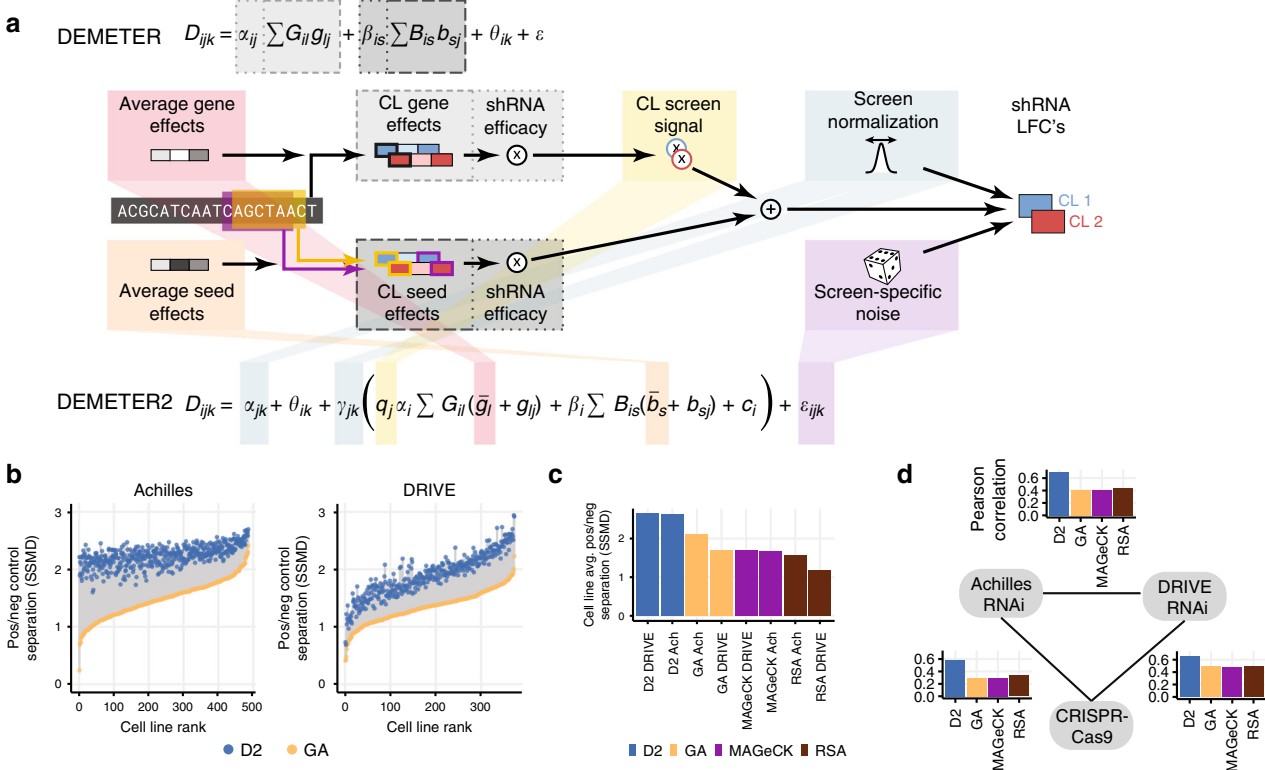

**Fig. 1** DEMETER2 improves identification of essential genes. **a** Both D1 and D2 represent the observed shRNA log fold change (LFC) depletion values in each cell line (CL) as a combination of gene knockdown and off-target seed effects. D2 introduces a number of additional model components highlighted in the schematic diagram. **b** Separation of gene dependency distributions for known common essential genes and non-essential (unexpressed) genes is measured by the strictly standardized mean difference (SSMD). Positive/negative control separation was much better for DEMETER2 gene dependency scores (blue dots) compared with per-gene averaging of shRNA depletion scores (GA; yellow dots) in both the Achilles (left) and DRIVE (right) datasets. **c** D2 estimates of across-cell-line average gene dependency showed improved separation of positive and negative control genes compared with previous methods. **d** Across-cell-line average gene dependency scores were in better agreement between datasets (Achilles RNAi, DRIVE RNAi, and CRISPR-Cas9 data) when using D2 estimates compared with previous methods. Each bar chart shows the correlation of average dependency scores between a pair of datasets. Colors represent agreement when using different models for estimating dependencies from RNAi data

lines tested; Fig. 1b). Less dramatic, though similarly consistent, improvements were also observed for the DRIVE dataset (SSMD increased by 42% on average, with improvement for all 373 cell lines tested). Furthermore, the improvements observed with D2 for both datasets were even larger when compared with the redundant siRNA activity (RSA) method[6], which was employed by the DRIVE study for identifying essential genes[2], as well when compared with MAGeCK[18], a method commonly applied to pooled screens with sequencing-based readouts (Supplementary Fig. 1).

The use of ~20 shRNAs per gene in the DRIVE dataset (compared to ~5 in Achilles) could ostensibly permit very effective extraction of the common on-target activities of same-gene shRNAs by average or RSA statistics. The robust improvement provided by D2 in this case highlights the benefits of model-based normalization, information-pooling across cell lines, as well as inference of shRNA efficacy, for the identification of essential genes. Perhaps surprisingly, positive/negative control separation was lower overall for the DRIVE dataset compared to the Achilles dataset (Supplementary Fig. 2a, b). This difference is likely due to the lower average on-target efficacy of shRNAs in the DRIVE library (Supplementary Fig. 2c), which is consistent with the necessarily less selective design criteria needed to create a library with 20 shRNAs per gene. The extra information provided by additional shRNAs per gene nonetheless reduces false positive signals and improves estimates of differential gene dependency across cell lines (Supplementary Fig. 2d, e).

We next evaluated the ability of these models to identify common essential genes, using the average dependency score across cell lines for each gene. For both the Achilles and DRIVE datasets, the application of D2 resulted in a much better separation of positive and negative control genes[17] when assessing the average dependency of each gene, compared with GA, RSA, and MAGeCK (Fig. 1c). As a further test of the accuracy of average gene dependency estimates, we compared them with estimates obtained in a genome-wide CRISPR-Cas9 screening dataset ($n = 391$ cell lines), using the CERES model to correct for gene-independent DNA-cutting toxicity effects[19]. For the Achilles data, D2 estimates showed a 2-fold increased correlation with CRISPR-based estimates compared with GA (Pearson $r$; D2 = 0.58; GA = 0.29), and there was a similar, though less pronounced, improvement for the DRIVE dataset (D2 = 0.65; GA = 0.49; Fig. 1d). Furthermore, the agreement between Achilles and DRIVE estimates of average gene dependency was much higher with D2 ($r = 0.70$) compared with GA ($r = 0.41$), RSA ($r = 0.44$), or MAGeCK ($r = 0.41$).

Thus, D2 addresses a key limitation of previous methods[1,8] by providing estimates of gene dependency on an absolute scale, allowing direct comparison across genes. Furthermore, D2 greatly improves identification of common-essential genes compared with existing approaches.

**D2 corrects screen-quality biases**. As shown in Fig. 1b, there were large differences in the quality of screening data across cell

lines (in terms of the separation of positive and negative control gene dependencies), in both the Achilles and DRIVE datasets. When using existing methods, cell lines with lower screen quality appear to be systematically less dependent on nearly all common essential genes, and conversely for cell lines with high screen quality (as illustrated for example low- and high-quality screens in Fig. 2a). Such global differences are likely due to assay-specific technical factors, rather than real differences in genetic dependencies. Indeed, differences in screen quality were associated with the expression of *AGO2* (Fig. 2b), the catalytic component of the RISC, suggesting they reflect variation in the efficacy of the underlying RNAi machinery across cell lines.

As demonstrated below, these differences can lead to substantial confounding effects in downstream analyses. To address this problem, D2 infers a "screen signal" parameter for each cell line, and effectively removes this source of bias from the estimated gene dependency scores. The model-inferred screen signal parameters are closely related to measured differences in screen quality (Supplementary Fig. 3a). They also show good agreement when estimated independently from the Achilles and DRIVE datasets (Supplementary Fig. 3b), suggesting that they capture robust differences in how different cell lines behave in RNAi screens. By estimating and accounting for these differences in screen signal, the gene dependency scores estimated by D2 for the same example cell lines no longer show systematic deviations from the across-cell-line average (Fig. 2a, bottom).

To show the magnitude of the effect of screen-quality related biases on estimated gene dependencies, as well as their successful removal with D2, we calculated the correlation across cell lines between *AGO2* mRNA expression and dependency scores for each gene. When using D1, the dependency profiles of many genes showed strong anticorrelation with *AGO2* expression, and the strength of anticorrelation was systematically increasing for genes that were more essential on average (Fig. 2c). In contrast, D2 dependency profiles showed little correlation with *AGO2* expression, even for common essential genes. These screen-quality related biases were not specific to D1, but rather were present to similar degrees using other methods, as well with both the Achilles and DRIVE datasets (Supplementary Fig. 4). While screen-quality related biases could in principle be removed by simply rescaling each cell line's gene dependency scores to better align data across cell lines for positive and negative control genes, this process dramatically magnifies the effects of noise in the lowest quality cell lines (Supplementary Fig. 5a). Similarly, we found that removing the first principle component of the D1 gene dependency matrix can provide effective correction of screen-related biases (Supplementary Fig. 5b, c). However, applying such post-hoc correction methods again produced poorer performance in downstream analyses (Supplementary Fig. 5d).

As an illustration of how screen-quality biases can impact downstream analyses, we computed correlations between the patterns of dependency across cell lines for each pair of genes. Such dependency correlation analyses have recently been shown to provide a powerful mechanism for identifying functional relationships as well as physical interactions among genes[1,2,20]. Since the screen-quality related biases are shared across genes, they can significantly influence estimates of dependency correlations, artificially inflating correlations between common essential genes. Indeed, we found that pairwise dependency correlations estimated using D1 increased systematically for gene pairs that were more pan-essential (Fig. 3a), and that this relationship was not particular to D1 (Supplementary Fig. 6). In contrast, dependency correlations estimated using D2 showed no such bias (Fig. 3b). To highlight how D2 can improve identification of functional interactions between genes, we consider an example

co-dependency network computed for the mediator complex gene *MED14* (see Methods). When using D1, *MED14* was connected with several other mediator complex genes, as expected. However, as *MED14* is essential in many cell lines it also connected strongly with a large group of common essential genes (Fig. 3c). When using D2, connectivity with the group of common essential genes was removed, and the co-dependency network of *MED14* was much more selective for other members of the mediator complex (Fig. 3d).

**D2 improves estimates of relative gene dependency.** Of particular interest to understanding cancer genetic dependencies is the ability to identify differences across cell lines, such as dependencies associated with a particular subtype of cancer, or with a particular biomarker. To assess the ability of D2 to accurately estimate relative differences in dependencies across cell lines, we compared the dependency profiles across cell lines for each gene with dependency estimates derived from the Achilles CRISPR-Cas9 dataset[19] for the same genes and cell lines. While there may be some differences in the consequences of shRNA-mediated vs. sgRNA-mediated effects, better agreement with CRISPR-based gene dependency profiles should reflect, in general, an improved ability to isolate and quantify on-target gene-knockdown effects. We found that D2-based estimates of gene dependency were in better agreement with those derived from CRISPR data, compared to previous methods, in both the Achilles (Fig. 4a) and DRIVE data (Supplementary Fig. 7a). As expected regardless of method, correlations between the RNAi and CRISPR dependency estimates were better for genes with stronger viability effects overall. Consistent with the greater impact of screen-quality related biases on essential genes (Fig. 2), the improvements in D2 compared with other methods were systematically larger for genes that were more essential on average.

We also tested the ability of different models to recover expected relationships between genetic dependencies and genomic features using several approaches (see Methods). First, we computed the correlation between each gene's dependency and its mRNA expression levels, as well as its relative copy number. D2-based dependency estimates showed a significantly stronger correlation with the gene's own expression (Fig. 4b), and copy number (Fig. 4c). In both cases, differences between D2 and other models were again most pronounced for genes that were more essential on average. These common essential genes typically show a strong positive correlation between dependency scores and the genes' copy number and expression levels, reflecting the CYCLOPS relationships[1,21], where partial loss of an essential gene renders a cell more sensitive to further suppression of the gene. As an example, we consider dependency profiles estimated for the common-essential ribosomal gene *RPL37*. When using D1, *RPL37* dependency is weakly correlated with *RPL37* copy number (Pearson's $r = 0.18$), instead showing a much stronger correlation with inter-cell line differences in screen quality ($r = -0.69$; Fig. 4d). In contrast, D2 largely removes the spurious association with screen signal ($r = 0.11$), resulting in an estimated *RPL37* dependency profile that is much more highly correlated with copy number variations ($r = 0.50$).

Finally, we tested the ability of D2 to recapitulate the most robust relationships between genetic dependencies and genomic features (dependency-feature pairs) identified from the CRISPR-Cas9 dataset (see Methods). When using D2, we found significantly stronger correlations between these dependency-feature pairs compared with D1 for both Achilles ($p = 2.6 \times 10^{-9}$, Wilcoxon signed rank test, $n = 384$ pairs; Fig. 4e) and DRIVE ($p = 5.4 \times 10^{-4}$, $n = 231$ pairs; Supplementary Fig. 7d) data. Furthermore, the improvements provided by D2 were even more

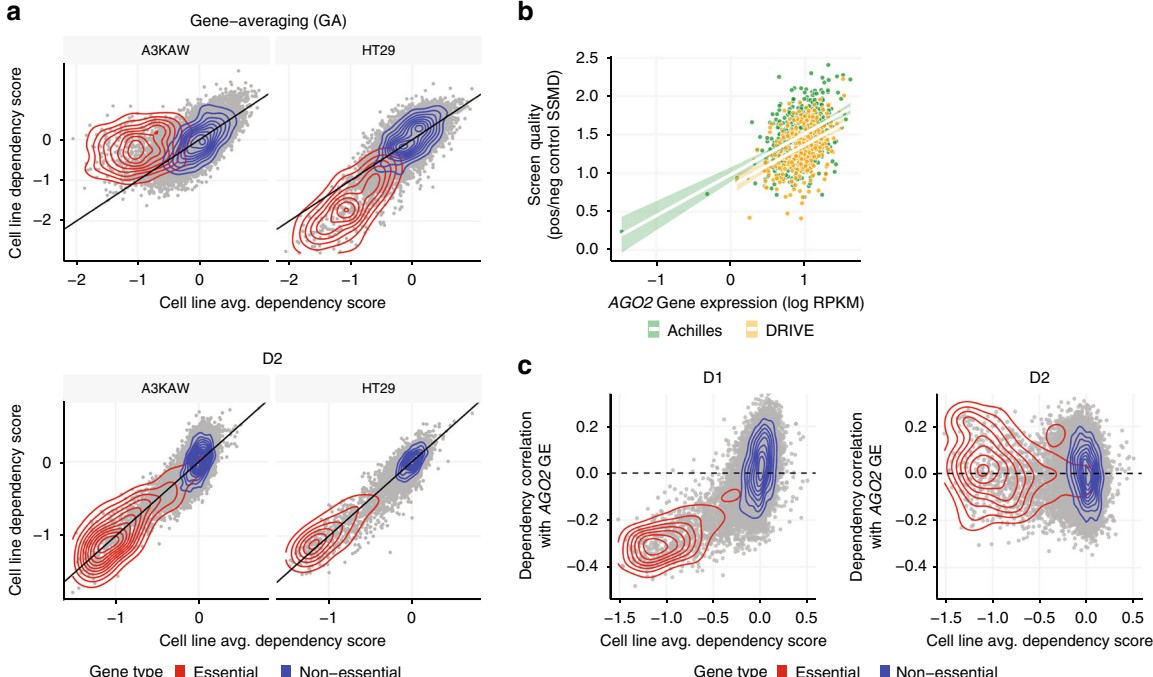

**Fig. 2** D2 corrects biases related to variable screen quality. **a** Comparison of across-cell-line average gene dependency scores with scores estimated for individual example low- (left) and high- (right) quality screens. Density estimates for the set of gold-standard common essential and non-essential genes are highlighted by the red and blue contours, respectively. Estimates using gene-averaging (GA; top plots) show broad systematic differences across all essential genes in these cell lines compared with the population average. These systematic differences are corrected for by D2 (bottom plots). **b** The screen quality estimated for each cell line (SSMD of positive/negative control gene dependencies, using GA) was correlated with the expression level of *AGO2* for both the Achilles (Spearman's rho = 0.39; $p < 2.2 \times 10^{-16}$; green) and DRIVE (rho = 0.37; $p = 1.3 \times 10^{-13}$; gold) datasets. **c** Correlation between each gene's dependency profile and mRNA expression of *AGO2* is plotted against the across-cell-line average dependency score for the gene, with curated common-essential and non-essential genes indicated with red and blue dots, respectively. Using D1 (left), gene dependency profiles were systematically (negatively) correlated with the *AGO2* expression for more common essential genes. This correlation was eliminated using D2 (right)

pronounced compared with other methods such as ATARiS, RSA, MAGeCK, and GA.

The ability of D2 to estimate the uncertainty of each gene dependency score allows for downstream analyses that account for the large differences in uncertainty across cell lines and genes. As expected, D2's estimates of dependency uncertainty were closely related to the screen quality and replicate agreement of each cell line, as well as differences in the number and quality of shRNAs targeting each gene (Supplementary Fig. 8a, c). In fact, the analyses presented here were all performed by weighting each gene dependency score by its associated precision (see Methods). Given the large differences in uncertainty across cell lines and genes, there is a good reason to expect improvements when using analyses that appropriately account for the uncertainty of each measurement. Indeed, we found that the agreement between RNAi and CRISPR-Cas9 dependency profiles decreased when we did not use gene dependency uncertainties to weight the correlation estimates (Supplementary Fig. 8d), but the D2-based results still provided improvements over other methods across all metrics, even when using uncertainty-agnostic methods.

These observations show that D2 improves estimates of relative differences in dependency across cell lines, allowing for improved identification of genomic features associated with genetic dependencies.

**Integration of multiple RNAi datasets**. The availability of the Achilles and DRIVE RNAi datasets, as well as other genome-scale RNAi screens, raises the possibility of integrating these data to provide a single comprehensive set of gene dependency estimates.

However, such data integration is challenging due to systematic differences between screens (batch effects), variable noise levels, and partial overlaps of the cell lines and shRNAs used. The ability of D2 to estimate batch-specific normalization factors and noise parameters in a statistically principled modeling framework make it well-suited to address these challenges.

We applied D2 to the combined Achilles and DRIVE datasets, along with a set of 76 breast cancer cell line genome-wide screens[3]. The combined dataset contains data from 974 RNAi screens of 712 unique cell lines across a broad range of cancer types, using 241k unique shRNAs (Fig. 5a). A primary concern when merging data across multiple datasets in this way is that large dataset-related differences (batch effects) will remain in the combined dependency estimates, confounding downstream analyses. Indeed, other methods of integrating across datasets— such as computing per-gene averages on the normalized and pooled data—produced results with strong screen-related batch effects (Fig. 5b). In contrast, these batch effects were greatly reduced when D2 was used to integrate multiple datasets. The problem of batch effects when combining datasets was largely specific to methods that estimate gene dependencies on an absolute scale (Supplementary Fig. 9a), presumably because methods that only assess relative dependencies per gene are less affected by, e.g., library-related differences between datasets.

We next sought to determine whether the integration of these RNAi datasets using D2 provided improved results due to the increased sample size and/or more accurate dependency estimates. Indeed, the combined D2 dataset showed small but consistent improvements in the dependency estimates for genes

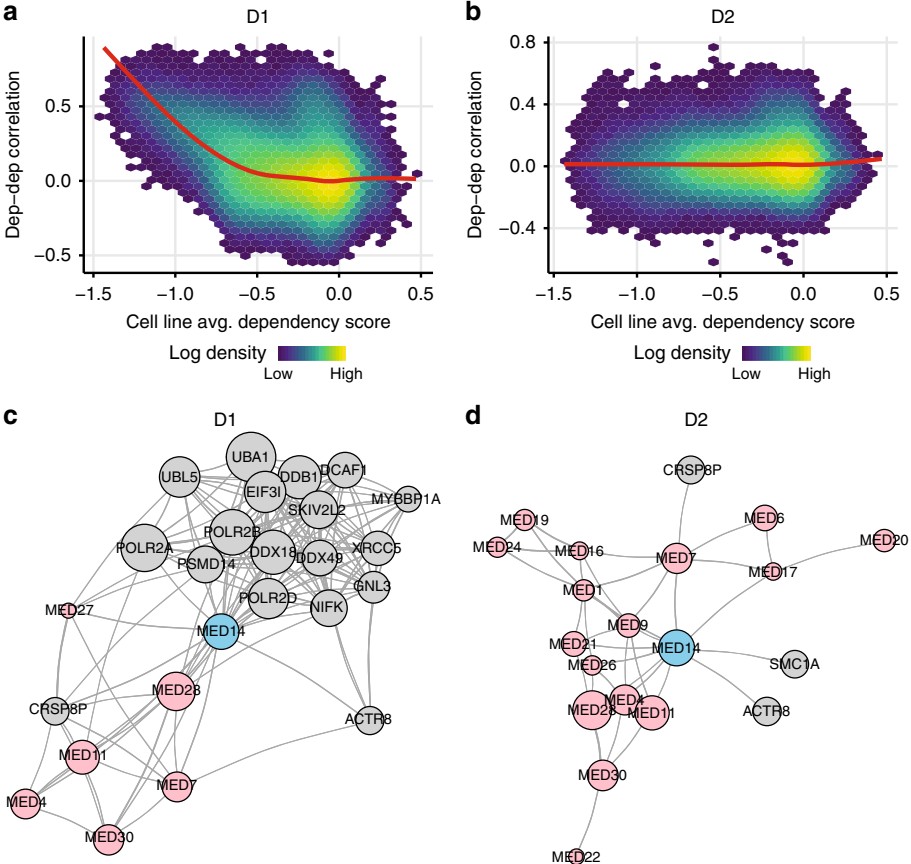

**Fig. 3** Screen quality biases impair dependency correlation analyses. **a** Correlation between pairs of gene dependency profiles, estimated using D1 applied to the Achilles data, increased systematically for gene pairs that were more essential on average. Red line shows the smoothed average. Color shows the density of data points in each region (log color scale). **b** Same as **a** but using D2, showing that the systematic upward bias in the pairwise correlation between pairs of common essential genes is removed. **c** Gene dependency correlation network surrounding *MED14*, constructed using D1 dependency estimates applied to the DRIVE dataset. Each node represents a gene, with edges depicting strong pairwise correlations (see Methods). Red nodes indicate genes that form protein complexes with *MED14*. Node size indicates the across-cell-line average dependency score for each gene (larger nodes representing more common-essential genes). **d** Same as **c** using D2, showing that the local dependency correlation network for *MED14* is more enriched for co-complex members

and cell lines screened in both Achilles and DRIVE, even though the total number of shRNAs targeting these genes was only marginally increased in the combined dataset compared with using the DRIVE dataset alone. For example, the combined D2 dataset showed better agreement with CRISPR-Cas9 estimates of per-gene average dependencies (Supplementary Fig. 9b), as well as dependency profiles across cell lines (Supplementary Fig. 9c). The most significant advantage offered by the integration of these RNAi datasets, however, is the increased coverage of cell lines, and the resulting increase in statistical power to identify relationships in the dependency data. For example, the same benchmark set of dependency/genomic feature relationships identified using CRISPR-Cas9 (as in Fig. 4e) were identified with much higher statistical significance using the combined D2 dataset compared with using the individual datasets (Fig. 5c).

As a further test of the power of the combined D2 dataset for identifying relationships between dependencies and genomic features, we performed a simple global analysis whereby we determined the genomic feature that was most strongly correlated to each gene's dependency profile (considering mRNA expression, copy number, and mutation features). We then asked whether the top correlated genomic feature was associated with the same gene as the dependency, or with a gene that was known

to be biologically related (considering physical interactions, CORUM protein complex membership, and sequence paralogues; see Methods), using such known biological relationships as a proxy for correctly identified dependency-feature relationships.

Overall, when using the combined D2 dataset the number of such dependency-feature relationships identified was 92 and 24% larger respectively compared with applying D2 to either the DRIVE or Achilles datasets alone (Fig. 5d). The number of relationships identified with the combined D2 dataset was 2.0–2.5-fold larger compared with applying either D1 or gene-averaging to the Achilles or DRIVE datasets. To better understand the source of these improvements, we categorized these dependency-feature relationships as "CYCLOPS", "oncogene expression", "oncogene mutation", "paralog loss", and "interacting protein" (Methods). The combined D2 dataset provided improvements in identifying nearly all categories of relationships compared with using D2 on the individual datasets (Fig. 5e).

Compared with the D1 datasets, the combined D2 model provided the most dramatic improvements in identifying CYCLOPS relationships (e.g., 3.7-fold and 4.4-fold increases respectively compared to D1 Achilles and DRIVE data), reflecting the fact that correction of screen-quality bias had the most substantial impact on common essential genes which tend to

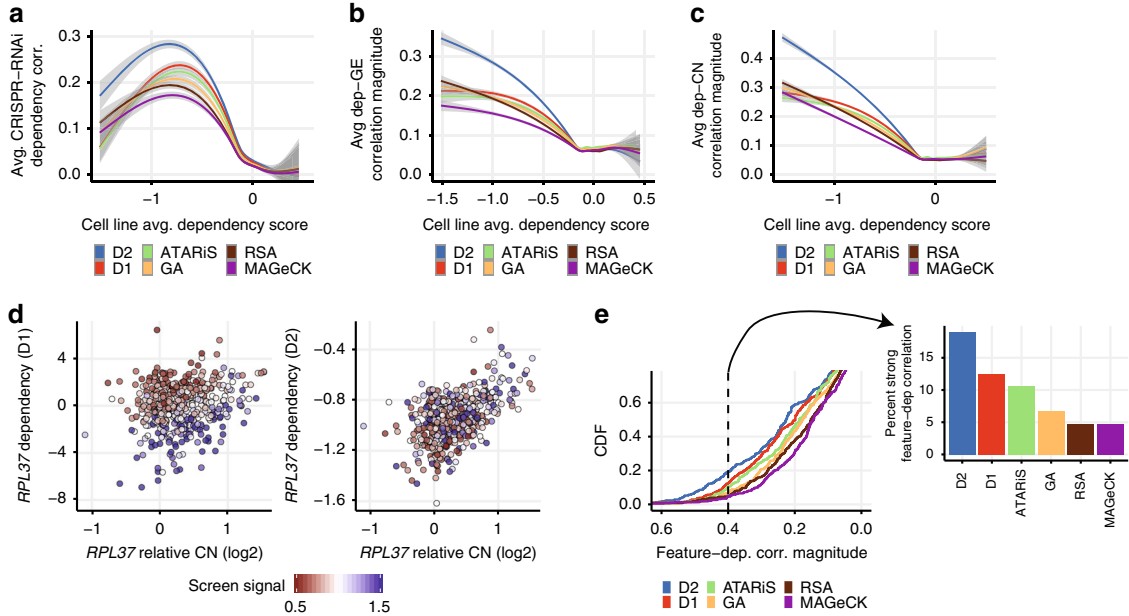

**Fig. 4** D2 improves estimated dependency profiles, particularly for essential genes. **a** Average correlation between RNAi and CRISPR-Cas9 gene dependency profiles as a function of the across-cell-line average dependency score, using the Achilles dataset. Different colored curves and shaded regions show the smoothed conditional mean correlation, and 95% confidence intervals, obtained using different models for estimating RNAi gene dependencies. D2 gene dependency estimates show better average agreement with CRISPR-Cas9 dependency profiles compared to existing methods. **b** Average magnitude of correlation between each gene's dependency and mRNA expression profiles (again for the Achilles dataset), plotted as a function of across-cell-line average gene dependency as in **a**. D2 dependency scores showed a stronger correlation with the gene's own expression levels compared with existing methods. **c** Similar to **b**, showing stronger correlations between D2 dependency profiles and the genes' own relative copy number, particularly for genes which are more essential on average. **d** Scatterplot of *RPL37* dependency vs. *RPL37* relative copy number using D1 (left) and D2 (right) dependency scores. Color represents the screen signal parameter estimated (from D2) for each cell line. **e** A benchmark set of dependency-genomic feature relationships identified from CRISPR-Cas9 data (see Methods) was used to evaluate the extent to which Achilles RNAi dependency estimates recapitulated the same associations. Colored curves show the empirical distributions of correlation magnitude across these dependency-feature pairs for each model. D2 dependency estimates showed better agreement with benchmark genomic feature associations compared to existing methods. Bar chart at the right shows the fraction of dependency-feature pairs with correlation magnitude greater than 0.4 for each model

show such relationships. To explore the improved identification of CYCLOPS-like relationships in more detail, we computed the correlation between gene dependency and gene dosage (using copy number and mRNA expression; see Methods) across all genes for both the combined D2 dataset, as well as the individual D1 datasets (taking the best correlation from either DRIVE or Achilles data for each gene). This analysis showed that the D2 combined dataset provided consistent increases in measured dose-dependency associations compared with D1 data for CYCLOPS-like genes (which was virtually all common essential genes; Fig. 5f). For some genes, such as *RPS29*, strong CYCLOPS relationships were detected in the combined D2 data, despite having only weak dose-dependency correlation in the D1 data. We also found substantial increases in the number of "paralog loss" relationships detected with the D2 combined data (2.1-fold and 2.8-fold increases compared to the D1 Achilles and D1 DRIVE datasets, respectively). A majority of the newly identified paralog loss relationships tended to have weaker correlation magnitude (Fig. 5g), suggesting they resulted from the increased statistical power of the D2 combined dataset. Taken together, these results show that the combined D2 dataset can substantially increase the utility of existing large-scale RNAi datasets for identifying genetic dependencies and their associated genomic features.

## Discussion

We present an improved model (DEMETER2) for inferring cancer cell line genetic dependencies from RNAi screens, and

show that it provides significant improvements over existing methods across a range of performance measures when applied to both the Broad Institute Achilles[1] and Novartis DRIVE[2] datasets. The D2 model also allows for effective data integration, and we apply it to combine three recently released RNAi screening datasets[1–3] to produce the largest compilation of cancer cell line genetic dependencies to date, comprised of 712 unique cell lines. We provide these data, along with the source code used to generate them, as a resource at https://depmap.org/R2-D2.

The predecessor of D2, DEMETER, was designed to address the strong off-target effects, and variable shRNA quality, which are well-known to confound interpretation of RNAi screening data. Here, we show that differences in screen quality between cell lines can pose additional challenges for efforts to map genetic vulnerabilities in cancer[1,2], by confounding comparisons of dependencies across cell lines. For example, when using existing methods, common essential genes appear to be systematically stronger dependencies in cell lines with higher screen quality, biasing downstream analyses such as estimation of gene–gene co-dependencies, and identification of molecular features predictive of dependency. D2 addresses this problem by incorporating explicit estimation of multiple screen normalization parameters from the data. Estimated "screen signal" parameters capture differences in the strength of gene suppression achieved in each cell line, and were remarkably reproducible when estimated from independent RNAi datasets. These parameters were also correlated with the expression of *AGO2*, a key component of the RNAi pathway, suggesting they reflect intrinsic differences in RNAi efficiency among cell lines[2,15,16]. Additional model-inferred

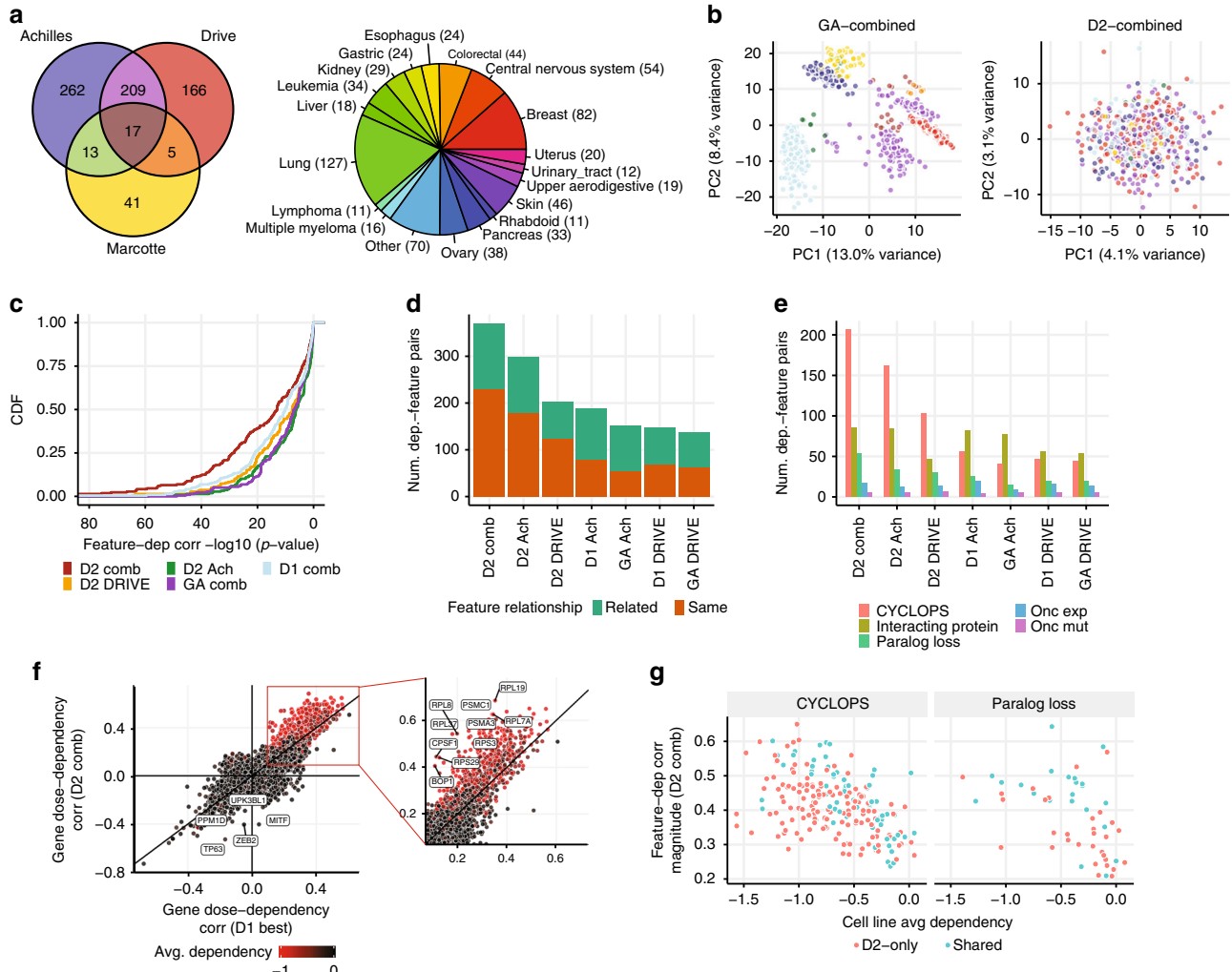

**Fig. 5** D2 effectively integrates multiple RNAi screen datasets. **a** Venn diagram showing the overlap of cell lines screened across the DRIVE, Achilles and Marcotte et al. datasets. Pie chart showing the composition of the combined dataset by primary disease. **b** The first two principal components of the gene dependency data for the combined GA (left) and D2 (right) data. Different colors represent which set of experiments were used to screen each cell line. Color scheme is the same as **a**, though light and dark blue indicate the cell lines screened with the "98k" and "55k" libraries in the Achilles dataset. **c** The statistical significance of measured associations between the benchmark dependency/feature pairs (same set as in Fig. 4e) is greatly increased when using the combined D2 dataset. Plot shows the empirical CDF of negative log p-values for dependency-feature associations using each model. **d** Number of gene's whose dependency profile is most correlated with a genomic feature that is from the same (red) or a related (green) gene, when using different models and datasets. **e** Dependency-feature associations identified by each model are classified as CYCLOPS, oncogene expression, oncogene mutation, paralog loss, and physical interactions. The D2 combined model identifies more relationships in nearly every class compared to using the individual datasets or other models. **f** Correlation between gene dependency and gene dosage (either mRNA or copy number) is compared across genes using either the combined D2 data or the best correlated of either D1 dataset (Achilles or DRIVE; see Methods). Subset at the right shows genes with strong positive dose-dependency correlations (i.e., CYCLOPS-like genes). **g** Comparison of feature-dependency correlation magnitude (in D2 combined data) with across-cell-line average dependency for CYCLOPS (left) and paralog loss (right) relationships. Color depicts which relationships were uniquely identified by the D2 combined data vs. shared with D1 datasets

normalization parameters captured inter-screen differences in the overall scale of shRNA depletion measurements, which were correlated with differences in the cell lines' measured growth rate (Supplementary Fig. 10). The D2 model thus identifies and removes multiple sources of cell line- and screen-related systematic bias in order to facilitate direct comparisons of genetic dependencies across cell lines. Surprisingly, the improvements gained by modeling such screen quality differences were often as large as those gained by accounting for off-target effects and variable shRNA efficacy (see, e.g., Figs. 4, 5, and Supplementary Fig. 7), which have been the focus of nearly all previous attempts to model RNAi screening data. Furthermore, simulations suggest

that D2 could provide robust improvements in performance when applied across a broad range of RNAi datasets, including with relatively few cell lines, and more focused libraries (Supplementary Fig. 12). This general approach—using a hierarchical modeling framework to pool information across cell lines, coupled with model-based normalization—could also be used to correct for similar sources of systematic bias in other functional screening assays, including CRISPR-Cas9 knockout screens.

Another limitation of previous models designed to address RNAi off-target effects[1,8] is that they only provide estimates of the relative differences in gene dependency across cell lines, precluding identification of common essential genes and direct

comparisons of dependency scores across genes. On the other hand, methods such as RSA[6] are more targeted towards calling essential genes in individual screens, such that multiple methods can be required to assess different aspects of RNAi screening data[2]. D2 addresses both of these use-cases by directly estimating gene dependency on an absolute scale that is comparable across genes and cell lines. Furthermore, we found that identification of essential genes was much improved using D2 compared to previous methods across all cell lines tested, and its estimates of across-cell-line average gene dependencies showed better agreement with curated common essential genes[17], as well as with CRISPR-Cas9-based estimates.

We previously processed the Achilles dataset of 501 RNAi screens with D1 to identify 769 genes of interest that show a strong differential dependency pattern across the cell lines[1]. As D1 generates only relative dependency scores, we used a cut-off of six global standard deviations from the mean to define this set ("six-sigma dependencies"). With the availability of absolute dependency scores from D2, more refined and stable approaches can be used to identify genes showing dependency patterns of interest. Notably, when we comparably analyzed the D2 Achilles dataset (using a threshold of 5.2 sigma to give an equal proportion of outlier genes) 57% of the previously reported genes that have D2 scores are re-identified. The main reasons for the discrepancy are the instability of the six-sigma metric used (a gene is called a six-sigma dependency even if it is a six-sigma dependency in a single cell line), and the screen quality bias-correction utilized by D2. We thus anticipate that the application of D2, and the availability of large loss-of-function datasets from CRISPR-Cas9 screens will permit further improvements in the identification of differential dependencies.

Finally, we note that the D2 model bears several similarities to previous methods that use hierarchical models. ScreenBEAM[22] uses a Bayesian hierarchical model to account for the variable efficacy across reagents targeting each gene. ScreenBEAM models each cell line independently, and is computationally expensive due to its use of Markov chain Monte Carlo (MCMC) sampling, and hence is less suited to large multi-screen datasets. The siMEM method[3] uses a mixed-effects modeling approach to account for differential gene dependencies across cell lines and variable shRNA efficacy. In contrast to D2, however, siMEM is designed to test associations between a given gene's dependency and a particular cell line property (e.g., comparing two groups of cell lines), treating differences between individual cell lines as random effects that are not directly estimated. D2 facilitates a broader range of possible analyses by providing accurate gene dependency estimates that can be directly compared with any given cell line property of interest (see, e.g., Figs. 4, 5). Some combination of existing methods might at least partially address the challenges described here, but we believe such approaches are unlikely to provide the full benefits of a unified statistical model such as D2. For instance, we explored several approaches for combining D1 dependency estimates that correct for off-target effects and variable shRNA efficacy with separate screen-quality bias correction procedures, but found that, unlike D2, they did not improve results in downstream analyses (Supplementary Fig. 5d), nor did they address other short-comings of these methods (e.g., lack of absolute dependency estimates).

In addition to improving gene dependency estimates compared with previous methods, D2 allows for an effective integration of data across multiple RNAi datasets. The resulting much-larger integrated RNAi dataset both improves the quality of gene dependency estimates, and also maximizes the coverage of cellular contexts and genes assayed, increasing the power of these datasets for discovering patterns in cancer cell genetic dependencies. We illustrate this by showing that the combined D2

dataset substantially increases the number of dependency-feature relationships identified compared with previous models, as well as with using the individual RNAi datasets. In summary, our results show that the combined RNAi dataset produced by D2 is a valuable resource that will greatly extend the utility of existing RNAi screening data.

## Methods

**Data processing**. For maximal consistency, we reprocessed raw shRNA read counts data for the DRIVE and Achilles datasets using the same pipeline. This consisted of first normalizing the counts data for each sample by computing the log counts per million, using the function "cpm" from the R package edgeR with a prior counts value of 10. Any shRNAs that did not have a log counts per million of at least 1 in the plasmid DNA were removed from the analysis. We then normalized each shRNA abundance by its associated value in the plasmid DNA sample to get log-fold change (LFC) estimates for each shRNA in each sample. These values were median-collapsed across replicates for each sample to get the LFC data serving as input to the models.

For Achilles data, the plasmid DNA measures were shared across samples within each of the three batches[1]. Hence, we used the replicate-collapsed plasmid abundances for each batch to estimate LFC values, as well as to identify shRNAs with insufficient plasmid representation, for all samples in the batch.

The raw DRIVE shRNA counts data (v4) were downloaded from https://data.mendeley.com/datasets/y3ds55n88r/4[23]. The cell line "f36p" was first removed from all analyses because it was a clear outlier in the number of sample read counts, with about 10 times the number of shRNAs exhibiting 0 counts compared to any other cell line. Any plasmid DNA measurement with insufficient counts was replaced by a "virtual library", calculated as described previously[2]. LFC values for shRNAs with the same targeting sequence in the same experiment were median-collapsed, along with any technical replicates, to obtain unique plasmid and sample counts for a given sequence in each pool and cell line.

For the Marcotte et al. dataset, the raw data were downloaded from the link provided in (http://neellab.github.io/bfg/; files used: "Normalized ExpressionSet", "updated shRNA annotations"). The file mapping probes to their sequences were downloaded from GEO (GSE74702. https://www.ncbi.nlm.nih.gov/geo/query/acc.cgi?acc=GPL21133). Probe sequences were mapped to their ids using the updated shRNA annotations file. LFC values were computed by taking the difference of $t_2$ and $t_0$ log2 measurements per replicate, and median-collapsing across replicates. For cases where a replicate had a $t_2$ measurement with no matching $t_0$ measurement, $t_0$ for this replicate was inferred by taking the mean $t_0$ measurement across all cell lines for that shRNA. We subsequently filtered out 8960 shRNAs that Marcotte et al. identified with a $t_0$ measurement below a noise threshold[1–3]. In addition, the cell line HCC1428 did not have $t_2$ measurements recorded and was therefore excluded from the analysis.

**Mapping shRNAs to genes**. Gene mappings were found by performing an exact string search through all RefSeq transcript RNA sequences (both protein-coding and non-coding) downloaded on June 28, 2017 from ftp://ftp.ncbi.nlm.nih.gov/refseq/H_sapiens/mRNA_Prot/human.*.rna.gbff.gz. For a given shRNA, the query sequence used in this search was the initial 19mer of the 21mer "target sequence". The final 1–2 bases of the 21mer tend to be cleaved off in the cell as an shRNA is processed into an siRNA, and thus they do not contribute to its targeting specificity. A shRNA was mapped to a gene if its initial 19mer was an exact match to *any* of the gene's transcripts in this search.

**The DEMETER2 model**. DEMETER2 (D2), as with DEMETER (D1), seeks to explain the observed shRNA depletion in each sample as a combination of gene knockdown effects and off-target seed effects. D2 expands the D1 model in several ways, as described below (and illustrated schematically in Fig. 1a). Let $D_{ijk}$ represent the depletion score measured for shRNA $i$ in cell line $j$ and dataset $k$, the D2 model is then given by the following equation (a complete description of the model parameters is given in Supplementary Table 1):

$$D_{ijk} = a_{jk} + \theta_{ik} + \gamma_{jk}\left( \begin{array}{c} q_j \alpha_i \sum_l G_{il}\left(\bar{g}_l + g_{lj}\right) \\ + \beta_i \sum_s B_{is}\left(\bar{b}_s + b_{sj}\right) + c_i \end{array} \right) + \epsilon_{ijk} \qquad (1)$$

In D2, the gene knockdown effect in a given cell line is explicitly modeled as a sum of two components: $\bar{g}_l$, the across-cell-line average effect for gene $l$, and $g_{lj}$ a component specific to cell line $j$. Similarly, the effects associated with seed sequence $s$ are represented by the across-cell-line average ($\bar{b}_s$) and cell-line specific ($b_{sj}$) effects of seed sequence $s$. This hierarchical model structure allows for information sharing across cell lines when estimating across-cell-line average gene and seed effects, while still effectively capturing inter-cell line variation. The set of genes $\{l\}$ and seeds $\{s\}$ targeted by a given shRNA are determined by the elements ($G_{il}$, $B_{is}$) of

fixed binary matrices that encode the shRNA-to-gene and shRNA-to-seed mappings.

As with the D1 model, each shRNA is assigned two seed sequences, given by positions 1–7 and 2–8 on the antisense strand. Hence, $B_{is} = 1$ if seed sequence $s$ appears in either of these seed regions of shRNA $i$. Similar to the D1 model, the efficacy of each shRNA in eliciting a given on- or off-target effect is modeled by parameters $\alpha_i$ and $\beta_i$ respectively, which are constrained to be in the unit interval [0,1]. Note that in the D1 model, separate efficacy parameters are estimated for each gene and seed targeted by a given shRNA. We found that these approaches gave very similar results, and using a single gene- and seed-efficacy parameter per shRNA allowed for better computational efficiency.

An important addition in D2 is the introduction of a "screen signal" parameter $q_j$ for each cell line, which scales how the cell line's gene effects are translated into shRNA-level depletion scores. This allows the model to account for global differences in the gene knockdown effects measured for different cell lines, such as arising from variable RNAi efficacy. Additionally, overall scale and offset parameters $\gamma_{jk}$ and $a_{jk}$ capture differences in the distribution of LFC values between screens. Since the model is invariant to global rescaling of the screen signal parameters and gene effects by a constant $a$ of the form:

$$q_j \to \frac{1}{a} q_j,$$
$$\left( \bar{g}_l + g_{lj} \right) \to a \left( \bar{g}_l + g_{lj} \right) \qquad (2)$$

and analogously for the overall scale terms, additional constraints are needed to ensure the identifiability of the model. Hence, we constrained both sets of scale parameters ($q_j$ and $\gamma_{jk}$) to have an average value of one, ensuring that they capture relative differences in scale across screens. Note that we assume that each cell line is characterized by a fixed screen signal parameter $q_j$ across datasets, while the parameters $\gamma_{jk}$ and $a_{jk}$ vary across different screens from a given cell line in order to capture batch effects.

Systematic shifts in the LFC values for an shRNA that are not captured by the gene and seed effect predictions, are modeled by additive components $\theta_{ik}$ and $c_i$. The former, which is a fixed offset across cell lines for each shRNA $i$ in dataset $k$, is designed to capture errors in the initial plasmid DNA measurements (which are shared across samples in the Achilles dataset). The $c_i$ on the other hand, are intended to model off-target effects not captured by the model-predicted seed-effects (shared across datasets using shRNA $i$).

Finally, the $\epsilon_{ijk}$ represent noise terms associated with each depletion measurement, which are assumed to be independently, normally distributed with screen-specific noise variance $\sigma_{jk}^2$.

**Parameter estimation**. We use a hybrid approach for parameter estimation, where Bayesian inference is used to estimate posterior distributions for the gene effects ($\bar{g}_l$, $g_{lj}$), seed effects ($\bar{b}_s$, $b_{sj}$), and intercept terms ($\theta_{ik}$, $c_i$, and $a_{jk}$), while point estimates (maximum a posteriori, MAP) are used for the remaining model parameters (the scale terms ($q_j$ and $\gamma_{jk}$), shRNA efficacies ($\alpha_i$ and $\beta_i$), and noise variances ($\sigma_{jk}^2$)). Initial point estimates of all parameters are constructed using an alternating block-wise coordinate ascent approach, after which a final stage of variational Bayesian inference is used to approximate the posterior distribution over $\Theta = \left\{ \bar{g}_l, g_{lj}, \bar{b}_s, b_{sj}, \theta_{ik}, c_i, a_{jk} \right\}$. The full procedure is outlined below:

(1) Initialize shRNA efficacy terms: $\alpha_i$, $\beta_i$, along with screen signal ($q_j$) and noise variance ($\sigma_{jk}^2$) parameters to 1.
(2) Initialize screen-specific scale terms $\gamma_{jk}$ by regressing the LFC data for each cell-line/batch on the average LFC across cell lines for that batch: $D_{ijk} = \hat{y}_{jk} \overline{D}_{ik} + c$.
(3) Estimate $\Theta$ given current estimates of remaining parameters.
(4) Estimate shRNA efficacies ($\alpha_i$, $\beta_i$) given current estimates of remaining parameters.
(5) Estimate screen signal parameters $q_j$ given current estimates of remaining parameters.
(6) Estimate overall scale parameters $\gamma_{jk}$ given current estimates of remaining parameters.
(7) Repeat steps (3)–(6) until convergence of the log-posterior.
(8) Initialize noise variance parameters $\sigma_{jk}^2$ by estimating the average residual variance of the model for each cell line/batch.
(9) Apply variational inference to estimate the posterior distribution of $\Theta$, along with the noise variances $\sigma_{jk}^2$, given point estimates of other parameters.

In steps 3, 4, and 6 we use SciPy's L-BFGS-B numerical optimization routine[24] to maximize the conditional posterior with respect to each parameter set. When estimating the shRNA efficacies we use bound constraints to ensure they are restricted to the interval [0,1]. To fit the overall scale parameters $\gamma_{jk}$, we maximized the posterior with the cell-line specific gene and seed effects ($g_{lj}$, $b_{sj}$) set to zero. This ensures that overall scale differences between samples were absorbed by $\gamma_{jk}$, rather than being incorporated in the estimates of $g_{lj}$ and $b_{sj}$.

The screen signal terms $q_j$ are updated by estimating the relative differences in measured gene effects for predefined positive and negative control gene sets. In particular, $\hat{q}_j$ are given by the difference between median positive control and

negative control gene effects:

$$\hat{q}_j = \left( \underset{l \in L_{neg}}{\text{median}} (\bar{g}_l + g_{lj}) - \underset{l \in L_{pos}}{\text{median}} (\bar{g}_l + g_{lj}) \right) \qquad (3)$$

We then normalize the $q_j$ to have an average value of 1 across cell lines. For the positive and negative control sets, we used the curated sets created by Hart et al.[17] We found largely similar results when updating the $q_j$ by maximizing the conditional posterior, as with the $\gamma_{jk}$ (and we provide this MAP estimation as an option in the open source version of the code). However, estimates obtained using positive/negative control gene separation provided more robust correction of systematic differences in screen quality between cell lines, particularly with the Achilles dataset. Note that while we use predefined sets of positive and negative control genes as part of the parameter estimation procedure, this does not create biases in the gene effect estimates for these genes. Rather, these gene sets are only used for estimating a global scaling of each cell line's gene effects relative to other cell lines, and hence do not affect, for instance, the rank order of gene effects for a given cell line. Furthermore, by using the medians across large gene sets (217/926 positive/negative control genes, respectively), the estimates of $q_j$ are insensitive to the inclusion of individual genes. Nevertheless, we performed cross-validation experiments where we split the positive and negative control gene sets into separate "train" and "test" sets to verify that this procedure does not introduce bias in our downstream model performance evaluation (Supplementary Fig. 11). For the final stage of model-fitting, we used a variational approximation to estimate the posterior distribution $p(\Theta|\mathbf{D}; \hat{\Psi})$, where $\Theta = \left\{ \bar{g}_l, g_{lj}, \bar{b}_s, b_{sj}, \theta_{ik}, c_i, a_{jk} \right\}$ is the set of parameters for which we estimate the posterior, $\mathbf{D}$ is the observed LFC data, and $\hat{\Psi}$ is the fixed vector of point estimates (MAP) for the remaining model parameters. We use a fully-factorized Gaussian (mean-field) model $q(\Theta; \lambda)$ to approximate the posterior, which is parameterized by $\lambda$: the set of marginal means and variances for each parameter in $\Theta$. The $\lambda$, along with the noise variances $\sigma_{jk}^2$ for each cell line/batch are then estimated by minimizing the KL-divergence $\text{KL}\left( q(\Theta; \lambda) | p\left( \Theta | \mathbf{D}; \sigma_{jk}^2, \hat{\Psi} \right) \right)$. To accomplish this, we utilized Edward[25], a probabilistic modeling language built on top of TensorFlow. In particular, we used the Edward function "KLqp", which uses stochastic variational expectation-maximization to simultaneously optimize $\lambda$ and $\sigma_{jk}^2$, by alternating between minimizing $\text{KL}(q|p)$ with given $\sigma_{jk}^2$ and maximizing $\mathbb{E}_{q(\Theta; \lambda)} \left[ p\left( \Theta, \mathbf{D}; \sigma_{jk}^2 | \Psi \right) \right]$ with respect to $\sigma_{jk}^2$.

**Simulation study**. To verify the effectiveness of this optimization procedure, we simulated data and tested the ability of the model to recover the known ground truth model parameters. In particular, we sampled shRNA LFC values for each cell line according to the conditional distribution specified by the DEMETER2 model, using parameters fit to the Achilles dataset. As shown in Supplementary Fig. 12a–d, the parameter optimization procedure described above was able to recover accurate estimates of the gene effect parameters and cell line scaling parameters with minimal bias. Errors in the point estimates of gene effects were also largely in line with their model-estimated uncertainties (Supplementary Fig. 12e), suggesting that the variational approximation of the posterior can provide reasonable uncertainty estimates.

We also performed simulations varying the number of cell lines, as well as the number of genes targeted by the shRNA library (again, simulating data based on the DEMETER2 model fit to the Achilles dataset, in this case using a subset of the Achilles data—the "98k" batch—for simplicity). These analyses showed that DEMETER2 can provide substantial improvements compared with simple gene-averaging, across several performance measures, even with a small number of cell lines, or a more focused shRNA library (Supplementary Fig. 12f–h). Nevertheless, the improvements provided by DEMETER2 over per-gene averaging were more pronounced when more cell lines were included, owing to fact that it pools information across cell lines, and accounts for differences in screen-quality.

**Priors and hyperparameter selection**. We use zero-mean Gaussian priors for the set of parameters $\Theta = \left\{ \bar{g}_l, g_{lj}, \bar{b}_s, b_{sj}, \theta_{ik}, c_i, a_{jk} \right\}$ for which we estimate approximate posteriors. For the remaining parameters, we assume uniform priors. The model thus uses hyperparameters that specify the prior variance associated with each parameter in $\Theta$ : $\sigma_{\bar{g}}^2$, $\sigma_g^2$, $\sigma_{\bar{b}}^2$, $\sigma_b^2$, $\sigma_\theta^2$, $\sigma_c^2$, and $\sigma_a^2$. In general, the results of the model were largely robust towards the precise choices of these hyperparameters (Supplementary Fig. 13). To select values for the prior variances $\sigma_{\bar{g}}^2$, $\sigma_c^2$, and $\sigma_\theta^2$ we performed a coarse grid search, choosing the values that produced average gene dependency estimates $\bar{g}_l$ with maximal separation between positive and negative control gene sets. For $a_{jk}$ we used an uninformative prior, setting $\sigma_a^2$ to an arbitrary large value.

We then performed a second grid search over values of $\sigma_g^2$ and $\sigma_b^2$, the hyperparameters controlling regularization of the cell-line specific gene and seed effects, by far the most numerous model parameters. While hyperparameters are often selected so as to minimize prediction error on held-out test data, we found that this approach resulted in over-regularization of the per-cell-line gene effects $g_{lj}$,

producing models that predicted little deviation from the mean gene effect $\bar{g}_l$ across cell lines (Supplementary Fig. 13). This is likely due to there being little variation in dependency across cell lines for a large majority of genes, and hence a model that strongly regularizes individual gene scores towards the across-cell-line average can perform best in terms of minimizing overall prediction error. Since the goal of the model is not to predict depletion levels of shRNAs in new experiments per se, but rather to extract the most biologically meaningful information about the gene knockdown effects $g_{lj}$, we instead coarsely selected values of the hyperparameters $\sigma_g^2$ and $\sigma_b^2$ based on a variety of measures, including agreement with CRISPR data, correlation with expected biomarkers, and the proportion of variance in the estimated gene effects attributed to between-gene vs. within-gene differences. Importantly, all the main results presented here were insensitive to the precise selection of hyperparameters, and we provide the values used for all hyperparameters in Supplementary Table 1 for reference.

**Data preprocessing**. As in the original DEMETER model, we exclude data for shRNAs that target more than 10 genes from the analysis, as such "promiscuous" shRNAs are likely to provide unreliable data. We also identified groups of genes that were targeted by identical sets of shRNAs. Since the models cannot distinguish the effects of knocking down individual genes within such groups, we combined them into single entities ("gene families") when estimating either DEMETER or DEMETER2 models[1]. For GA, RSA, ATARiS, and DEMETER, LFC values were z-score normalized per cell line and batch before model fitting. For the Achilles data, there were three different batches of cell lines screened, reflecting changes in the library and experimental methods[1]. For the DRIVE data, each cell line was screened using three shRNA libraries[2], creating three batches of shRNA data per cell line.

**DEMETER**. The DEMETER model was fit using the R source code provided at: https://github.com/cancerdatasci/demeter. Achilles LFC data were provided as input in three separate batches of cell lines[1]. For the DRIVE data, LFC values from different pools were all combined into a single matrix as input. shRNAs from different pools were considered distinct, even if they shared the same targeting sequence. The following parameters were determined by performing separate hyperparameter searches on the DRIVE and Achilles data (Table 1).

When applying DEMETER to the Marcotte et al. dataset we used the same hyperparameters as used for analyzing the Achilles dataset. Note that while DEMETER was previously applied to quantile-normalized LFC data[1], here we used z-score normalization to make the results more directly comparable to those produced by DEMETER2.

**ATARiS**. ATARiS was run using Gene Pattern (http://software.broadinstitute.org/cancer/software/genepattern/modules/docs/ATARiS/1) with the default parameters. The first solution for each gene was taken. The ATARiS algorithm was run separately on the "98k" and "55k" Achilles data. Note that we combined the two "55k" Achilles batches for running ATARiS, because they used nearly identical shRNA libraries, and one of the batches had too few cell lines to get reliable results.

**RSA**. RSA was run using the R implementation provided by the Genomics Institute of the Novartis Research Foundation (http://winzeler.ucsd.edu/supplemental/KonigNatureMethod-2007/RSA.html) with the following parameters:

- No Bonferroni correction
- Not reversed
- Lower bound: −1000
- Upper bound: 1000

The bounds were set to extreme values so that no gene would automatically be considered a hit, and genes targeted by only one hairpin were removed. Data were combined across batches for each cell line (after z-score normalization). Separate input files were created for each cell line and fed into the RSA algorithm.

**MAGeCK**. MAGeCK was run using the command-line tool (version 0.5.7), downloaded from https://sourceforge.net/p/mageck/wiki/Home/. We used the function "mageck mle" to compare late time point and early time point read counts data, using default parameters with the additional options: "--no-permutation-by-group", "--permutation-round 0", and "--norm-method none". Before running MAGeCK, we

first normalized for differences in library size such that the total read counts for each sample were equal to the median of the total read counts across all samples. For Achilles data, read counts for each replicate were provided as separate inputs.

**Processing gene dependency estimates**. Gene dependency scores for MAGeCK, RSA, GA, and D2 were normalized using a uniform scaling and offset (applied to all cell lines) so that the median of the across-cell-line average dependency scores for positive and negative control gene sets[17] were set to −1 and 0, respectively. Such normalization could not be applied with ATARiS and D1, which estimate gene dependencies on a relative scale. Hence, for these models, we applied a global z-score normalization of the dependency scores (mean-subtracting per gene, then normalizing by the global standard deviation).

For some genes the estimated dependency scores were deemed unreliable, and were excluded from all analysis. In particular, any genes that were targeted by fewer than three shRNAs were excluded. We also excluded genes that were determined (by the DEMETER2 model) to have poor quality reagents. Specifically, we removed genes where the average gene-knockdown efficacy ($\alpha_i$) was less than a minimum value (0.2), or where the sum of $\alpha_i$ across targeting shRNAs was less than a threshold of 1.5. These criteria resulted in the removal of 568/8393 genes for the DRIVE dataset and 358/16855 genes for the Achilles dataset. Finally, genes that were part of a "gene-family", sharing identical sets of targeting shRNAs, were excluded from the analyses, since it is not possible to distinguish the specific gene knockdown effects among genes within such a group[1]. For comparisons across models, we analyze only those genes for which we obtained valid gene effect estimates with D2, based on the above criteria, to ensure fair comparisons.

**Genomic features**. Gene expression data were taken from the file: CCLE_-DepMap_18Q1_RNAseq_RPKM_20180214.gct, downloaded from the Cancer Cell Line Encyclopedia (CCLE) portal (https://portals.broadinstitute.org/ccle/data). These RPKM values were then transformed according to log10(RPKM + 0.001), and our analysis was restricted to protein coding genes only. For identifying genes that were "unexpressed" in a given cell line, we utilized a log10(RPKM) threshold of −1.

Gene-level relative copy number data were derived from a combination of CCLE whole-exome sequencing (WES) and SNP data (https://portals.broadinstitute.org/ccle/data). We utilized WES and SNP data to achieve maximal coverage across cell lines. When multiple datasets were available for a given cell line, we prioritized WES over SNP data. Relative copy number data were also log-transformed for analysis.

For mutation data, we utilized the merged mutation calls file from the CCLE portal (CCLE_DepMap_18Q1_maf_20180207.txt), which combines information from multiple data sources and types. We considered mutations to be "damaging" if they were marked as "deleterious" in the maf file. A subset of missense mutations was further categorized as "hotspot" mutations, if they were annotated as being either TCGA hotspots or COSMIC hotspots.

**Dependency-feature relationship analysis**. We benchmark the ability of different models to identify relationships between genetic dependencies and genomic features in several ways. First, we used CRISPR-Cas9 gene dependency estimates, based on the CERES algorithm[19], to identify a set of top dependency-feature relationships which we could then test in the RNAi datasets using different models. In particular, for each CRISPR-Cas9 gene dependency profile we identified the most strongly correlated feature (based on the Pearson correlation) for each of four feature types (mRNA expression, relative copy number, damaging mutation, and hotspot missense mutation). We then took the top feature-dependency correlations for each feature type (up to 200 pairs per feature type), after removing any relationships that did not have a minimum correlation magnitude of 0.4, producing a set of 417 benchmark feature-dependency relationships (193 copy number, 200 gene expression, 11 damaging mutation, 13 hotspot missense mutation).

We also employed a list of known gene–gene relationships to test how frequently the genomic feature most correlated with a gene's dependency was from a gene known a-priori to have some relationship with the targeted gene. We used gene–gene relationships defined in several ways:

- Physical interactions: gene pairs that were identified as CORUM protein complex co-members[26], or as physically interacting using protein–protein interaction data from InWeb[27].
- Paralogs were defined as gene pairs which last underwent a duplication event rather than a speciation event according to Ensembl.

For the analysis in Fig. 5e, dependency-feature relationships were classified into the following groups:

- CYCLOPS: defined as cases where the top correlated genomic feature was either mRNA expression or copy number of the target gene, and the correlation was positive.
- Oncogene mutation: defined as cases where the top correlated genomic feature was the gene's own hotspot missense mutation status, and the correlation was negative (stronger dependency in the mutant cell lines).
- Oncogene expression: defined as cases where the top correlated feature was the gene's own mRNA expression, and the sign of the correlation was negative.

| Table 1 Parameters | | |
| --- | --- | --- |
| | **Achilles** | **DRIVE** |
| **randseed** | 1 | 1 |
| **G.S** | 1.67e-5 | 2.38e-4 |
| **alpha.beta.gamma** | 0.583 | 0.033 |
| **max.num.iter** | 500 | 500 |
| **learning.rate** | 0.005 | 5e-5 |

- Paralog loss: defined as cases where the top correlated feature was either damaging mutation, copy number, or gene expression of a sequence paralog to the target gene, and the correlation was positive.
- Interacting protein: defined as cases where the top correlated feature was from any gene known to physically interact with the target gene (using either positive or negative correlations).

Dependency-feature relationships that fit multiple of the above categories were prioritized in the order described above (e.g., relationships would only be classified as "physical interactors" if they did not meet the criteria for any of the other categories).

**Additional analysis details**. Wherever possible we utilized weighted statistics (including Pearson correlations, means, and variances) to evaluate the quality of D2 gene dependency estimates, where dependency scores were linearly weighted by their associated precision (the inverse posterior variance). Weighted Pearson correlations, and associated $p$-values, were computed using the R package "weights" (https://CRAN.R-project.org/package=weights).

For PCA analysis (Fig. 5b), we used probabilistic PCA, implemented in the R package pcaMethods[28], which naturally handles missing values in the gene dependency matrices. PCA was applied to the matrices of dependency estimates from each model after mean-subtracting per gene.

Co-dependency network visualizations (Fig. 3c, d) were generated using the R package igraph[29]. We first identified the top 25 gene dependency profiles most strongly correlated (magnitude of Pearson correlation) with the query gene's dependency profile, using these genes as the nodes of the graph. Edge weights between nodes were given by the magnitude of Pearson correlations between gene pairs, using only edges where the correlation magnitude was >4 $z$-score above the mean (across all gene pairs for each dataset). Disconnected nodes were then trimmed from the graph before generating the plots using the "layout_nicely" algorithm in igraph.

To generate Fig. 3a, b, as well as Supplementary Figure 6, we computed pairwise dependency correlations using a subset of genes for computational efficiency. Specifically, we used the top 2500 genes with highest across-cell-line variance (according to DEMETER), considering only genes present in both Achilles and DRIVE.

Paired two-sample comparisons were made using Wilcoxon signed rank tests. The significance of Pearson correlations was assessed using the R functions cor.test (with the function wtd.cor from the R package weights used for precision-weighted correlations). For computing dependency-feature association $p$-values (Fig. 5c), we used the Pearson correlation $p$-value for continuous features (gene expression, copy number), and $p$-values from two-sample $t$-tests for binary features (mutations). Two-sided $p$-values were used in all cases. Figures showing conditional average correlations (with shaded uncertainty regions) were created using the "geom_smooth" function from the R package ggplot2[30].

For the analysis in Fig. 5f, we computed dose-dependency correlations by correlating each gene's dependency profile with its own copy number and mRNA expression across cell lines. The correlation with the larger magnitude was taken as the gene's dose-dependency correlation (for D2, we again used precision-weighted correlations).

**Code availability**. The full source code implementing the model, data pre-processing, and figure generation is made available at https://github.com/cancerdatasci/demeter2.

## Data availability
All datasets used to generate the results presented here are publicly available. The results of the DEMETER2 model applied to the DRIVE and Achilles datasets, as well as to the combined DRIVE, Achilles and Marcotte et al. data, are available at https://depmap.org/R2-D2, and in a Figshare record at: https://doi.org/10.6084/m9.figshare.6025238.v4. This Figshare record also includes the log-fold change data and other inputs to the DEMETER2 model, as well as cell line copy number, gene expression, and mutation datasets used in the analysis. CRISPR-Cas9 essentiality screening data processed using the CERES algorithm can be downloaded from https://doi.org/10.6084/m9.figshare.5863776.v1[31]. All other remaining data are available within the Article and Supplementary Files, or available from the authors upon request.

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

## Acknowledgements
This work was funded in part by the Carlos Slim Foundation. We thank Andrew Tang for help with the figure design.

## Author contributions
J.M.M. and A.T. conceived and designed the study. J.M.M. wrote and implemented the modeling software. J.M.M. and Z.V.H. performed computational analysis and interpretation of results. G.K., J.M.D., P.G.M., J.G.B., J.M.K-B., and T.M.G. assisted with computational analysis and model design. P.G.M. provided computational tools. J.M.M.,

Z.H., D.E.R., and A.T. wrote the manuscript. F.V., J.S.B., T.R.G., W.C.H., D.E.R., and A.T. reviewed and revised the manuscript. A.T. supervised the study.

## Additional information

**Competing interests:** W.C.H. is a consultant for Thermo-Fisher, Paraxel, AjuIB, MPM Capital and KSQ Therapeutics and receives research funding from Deerfield Management. W.C.H. is a founder and has equity in KSQ Therapeutics. T.R.G. is a consultant to Foundation Medicine and GlaxoSmithKline, and is a shareholder of FORMA Therapeutics. D.E.R. receives research funding from members of the Functional Genomics Consortium (Abbvie, Jannsen, Merck, Vir), and is a director of Addgene, Inc. A.T. is a consultant for Tango Therapeutics. All remaining authors declare no competing interests.

