## [Peer Review File · Nature Communications]

Reviewers' Comments:

Reviewer #1:

Remarks to the Author:

McFarland et al. introduce DEMETER2, a hierarchical model pooling information across cell lines, coupled with model-based normalization, to correct for systematic bias seen in large perturbation screens. DEMETER2 (D2) is an extension of DEMETER (D1) and estimates absolute gene dependency scores by correcting for off-target effects, variable shRNA quality, and screen quality among cell lines, thus enabling effective integration of multiple RNAi datasets. They show on three different screens that D2 provides an improved absolute gene dependency score in comparison to existing methods like gene averaging (GA), and the redundant siRNA activity method (RSA). The main novelty lies in explicitly modeling the screen quality, thereby correcting for systematic dependencies associated with screen quality that bias the downstream analyses.

The model has several parameters, which are estimated using a hybrid approach based on variational inference. Two potential issues that require further explanations are overfitting the data and identifiability of the model. The authors only mention that the multiplicative scaling parameters are constrained to an average of 1 for the model to be identifiable, but do not provide any reasons. The authors should further elaborate on this choice, why it makes the model identifiable and why overfitting is not an issue. The authors should address these and other issues like when the method breaks down in a simulation study with varying parameters.

Additionally the authors should compare D2 to existing methods like siMEM despite having argued against it, as the explanation is not very convincing. Alternatively the biases can also be addressed by applying several existing methods in combination, e.g. first getting rid of off-target effects and then correcting for screen/cell lines bias. Several methods for both steps exist and have been mentioned by the authors. They should compare D2 to such approaches that subsequently apply methods for the two subtasks, i.e. off-target correction and integration across screens, both in simulations and when applied to the real data.

The authors describe in detail how the binary shRNA-to-gene mapping G is computed. They should also describe how to compute the shRNA-to-seed mapping B. The authors do not mention that and the method used for G does not seem to be transferable to B.

Figure S8c is missing a color key for the colors depicting the density of points.

On page 25 the sentence fragment "the D1 model in several ways, as described below (and illustrated schematically in Fig." was accidentally copy and pasted in the next sentence and appears twice.

In general, the paper is well written, sound, and relevant to the perturbation community. However, we cannot recommend the paper for publication before the authors address the above concerns, especially on simulations and comparisons to other approaches.

Reviewer #2:

Remarks to the Author:

McFarland et al describe a new algorithm, DEMETER2, for the analysis of RNAi (and CRISPR) screens. Significant resources have been invested in the shRNA-based pooled library screening of cancer cell lines to identify "synthetic lethal" targets for therapeutic intervention. Even with the substantial advantage that CRISPR screening brings to bear on this problem, efforts to improve the analysis of the very large number of shRNA screens available are well justified.

The authors present a comprehensive upgrade to DEMETER that includes parameters for overall

screen quality (itself estimated by evaluating the screen's ability to separate reference sets of essential and nonessential/nonexpressed genes), variable gene-level and seed-level (i.e. off-target) effects, and provides an estimate of absolute fitness defect for each gene. All of these are critical improvements that contribute to the impact of this algorithm.

Overall this study represents an important contribution to the field. It identifies concepts that are key weaknesses in the existing literature/methods, addresses them in a statistically robust manner, and potentially rescues a lot of data from oblivion.

Minor edit:

Fig 2b: the use of Spearman correlations may render this point moot, but the robustness of the Achilles correlation should be evaluated with regard to the removal of the two outliers (screens w/AGO2 expression < 0).

Reviewer #3:

Remarks to the Author:

Summary: The authors have developed Demeter2, a significantly improved implementation of their previous method Demeter, to correct for technical and batch artifacts in RNAi screens of cancer cell lines. Despite some inherent limitations, cell line-based RNAi screens provide an extremely valuable resource to the community for understanding genetic interactions and their relevance to cancer. Thus strategies to aggregate and denoise these data are of great interest and value to the broader biomedical research community. Reducing signal due to off target effects and other technical noise that contaminates such datasets is a challenging problem. The authors convincingly demonstrate that Demeter2 improves the quality of RNAi screen data beyond what was possible with other existing methods. While this work builds on a previously published tool, the modifications result in a large overall improvement and allow novel quantification of gene dependencies relative to the previous incarnation. Overall the manuscript is clearly written and represents an important advance, and importantly both the source code and data will be made available to the research community. This reviewer has only a few suggestions.

Major comments:

It is exciting to see that Demeter2 improves over other available methods. After convincingly establishing the improvement in performance, the authors spend some time showing how different genomic features and mechanisms correspond to gene dependencies. The majority of improvement seems to come with more confident identification of CYCLOPs effects. The opportunity exists here to go into more depth in characterizing the gene dependencies that are recovered by the Demeter2 approach. Is it a set of genes that confer a smaller effect size of dependency, or that the same genes being detected in new cell lines? Are these genes mostly known essential genes, or are novel genes implicated? Is there any functional coherence across these smaller effect-size dependencies? While this manuscript is clearly more technical in spirit, any biological insights that might come from the ability to detect smaller effect sizes could broaden the interest in the manuscript.

Minor comments:

Given that using D2 to combine datasets allowed assessment of the improvement in performance relative to the amount of data gained, it would be interesting to investigate the point where performance due to added data saturates for genes with different magnitudes of dependency.

Figure S8d was confusing at first since it wasn't clear that D2 figures previously were always accounting for uncertainty. It might be more clear if the description were framed as D2's

performance being modestly but significantly reduced when precision is not used to weight dependency estimates.

Figure 5b – add variance explained to the axis by each PC, or in the caption, for the different PCA plots

Since the resource will be made publicly available, it would be helpful to include a level description of its composition, for example, an overview of the cell type composition of the cell lines.

Reviewer #1 (Remarks to the Author):

McFarland et al. introduce DEMETER2, a hierarchical model pooling information across cell lines, coupled with model-based normalization, to correct for systematic bias seen in large perturbation screens. DEMETER2 (D2) is an extension of DEMETER (D1) and estimates absolute gene dependency scores by correcting for off-target effects, variable shRNA quality, and screen quality among cell lines, thus enabling effective integration of multiple RNAi datasets. They show on three different screens that D2 provides an improved absolute gene dependency score in comparison to existing methods like gene averaging (GA), and the redundant siRNA activity method (RSA). The main novelty lies in explicitly modeling the screen quality, thereby correcting for systematic dependencies associated with screen quality that bias the downstream analyses.

The model has several parameters, which are estimated using a hybrid approach based on variational inference. Two potential issues that require further explanations are overfitting the data and identifiability of the model. The authors only mention that the multiplicative scaling parameters are constrained to an average of 1 for the model to be identifiable, but do not provide any reasons. The authors should further elaborate on this choice, why it makes the model identifiable and why overfitting is not an issue. The authors should address these and other issues like when the method breaks down in a simulation study with varying parameters.

We thank the reviewer for highlighting these areas where additional explanation and validation of model-fitting details were needed.

Identifiability: We have revised the manuscript (Methods section DEMETER2: Model description; pg. 28) to better explain this point. In summary, the model is invariant to global scaling of the gene effects by α , if the multiplicative terms are all scaled by $1/\alpha$. Thus, additional constraints (one for each set of multiplicative scaling terms) are needed to ensure the identifiability of the model. We chose to constrain the multiplicative scaling terms to have an average of 1 so that they capture relative differences in normalization/screen-signal across cell lines, making the gene effects estimated by the model more directly comparable across cell lines.

Overfitting: We agree with the reviewer that this is an important issue, and we have added a supplementary figure (Fig. S13), and additional discussion to the Methods (section “Priors and hyperparameter selection”, pg. 32-33), to clarify our procedure for selecting hyperparameters. Overfitting of the model to training data is typically assessed by evaluating the accuracy of model predictions on held-out test data, which we did. The merits of a strategy of determining hyperparameter values based on the model’s prediction accuracy deserves special mention in this particular application. The purpose of our model is not to predict depletion values of shRNAs in pooled screens *per se*, but rather to extract biologically meaningful information from such screens. While we did measure the model’s prediction error on both train and test data, we found that refining hyperparameters by minimizing test error, as is commonly done, produced models where the gene effects were overly biased towards the across-cell-line average. This makes sense intuitively, as there is likely only modest real variation in dependency across cell lines for a large fraction of genes, and hence a model that strongly regularizes individual gene scores towards the across-cell-line average can perform best in this global sense. Nevertheless, this can create significant bias towards underestimating the across-cell-line variability for genes that are strongly differentially dependent across cell lines, which are often the genes cancer researchers are most interested in studying. As such, our strategy for selecting model hyperparameters was to also utilize other measures of model performance, such as separation of positive and negative control genes, and agreement of estimated gene effects with CRISPR-Cas9 data (though we note that such performance measures were largely insensitive to the precise selection of model hyperparameters). We also note that the improved performance we observe with DEMETER2 across a wide-range of downstream benchmark analyses suggests it is not strongly impacted by overfitting (which would tend to adversely affect these performance measures).

Simulations: We appreciate and adopted the reviewer's suggestion to use simulated data to validate the parameter estimation procedure we employ, as well as to explore the applicability of the model across different types of datasets. We now show that D2 is indeed able to recover accurate estimates of key model parameters using realistic simulated data (with parameters matched to those measured in the Achilles dataset). We added text to the methods (DEMETER2: Parameter estimation; pg. 31-32), as well as a supplementary figure (**Fig. S12**) describing the simulations. We note that the errors in the model's gene effect 'point' estimates are largely consistent with its uncertainty estimates (taken from the Gaussian-approximated posteriors).

We also used simulations to evaluate the performance of the model when applied to datasets with varying numbers of cell lines and genes targeted (described on pg. 32, **Fig. S12f-h**). These analyses show that the model can be applied robustly across a wide range of datasets, including with much smaller shRNA libraries, and with relatively few cell lines. As expected, however, the improvements provided by D2 were most pronounced when applied to datasets with more cell lines, due to the pooling of information across cell lines, and its accounting for screen-quality differences across cell lines.

Additionally the authors should compare D2 to existing methods like siMEM despite having argued against it, as the explanation is not very convincing. Alternatively the biases can also be addressed by applying several existing methods in combination, e.g. first getting rid of off-target effects and then correcting for screen/cell lines bias. Several methods for both steps exist and have been mentioned by the authors. They should compare D2 to such approaches that subsequently apply methods for the two subtasks, i.e. off-target correction and integration across screens, both in simulations and when applied to the real data.

siMEM: We agree that our explanation for why we did not compare our results with siMEM was unclear. We have now clarified this point in the revised manuscript (Discussion section, pg. 23). In siMEM, differences in gene effects across individual cell lines are treated as 'random effects', and hence are not directly estimated by the model. Instead, siMEM is designed to test for differences in gene effects only between groups of cell lines (or differences related to a continuous cell line parameter).

Additional models: In order to provide additional comparisons between DEMETER2 and existing methods, we ran MAGeCK on all cell lines in the Achilles and DRIVE datasets. MAGeCK is a popular method for estimating gene dependencies in pooled screens that is specifically designed to model the noise properties of sequencing-based readouts. Nevertheless, we found that MAGeCK did not perform better than simple gene-averaging on our evaluation metrics (see revised **Figs. 1, 4, S7**), potentially because it is designed for calling dependencies in individual screens, rather than extracting comparable measures of gene dependency across many screens. We also explored the use of ScreenBEAM, another hierarchical Bayesian model of pooled functional screens, but we found that it was computationally infeasible to apply to such large multi-screen datasets owing to its reliance on MCMC sampling (and we have made mention of this in the revised manuscript, Discussion section, pg. 23). It's unlikely that ScreenBEAM (or other previously described methods) will address the main challenges described here with large-scale RNAi screens, since it does not model RNAi off-target effects specifically, and treats data from each cell line independently.

Applying multi-stage methods:

We agree with the reviewer's point that the screen-quality related biases could be addressed using separate 'post-hoc' correction methods applied to gene dependency estimates obtained using existing methods. We thus considered such strategies in more detail in the revised manuscript (see pg. 11 in the main text, and Fig. S5), as summarized below.

In the original manuscript, we showed that rescaling each cell line's gene dependencies to align the positive and negative control genes' effects can eliminate the bias, although it introduces another problem by greatly magnifying the noise present in lower-quality screens. Based on the reviewer's comment, we explored additional approaches for 'post-hoc' correction of the screen-quality bias, and found that the first principle component of the gene dependency

matrix estimated using D1 can be used to capture the variation in screen signal across cell lines remarkably well (revised Fig. S5b). Hence, one approach would be to remove this PC from the D1 data. Indeed, we found that this approach performed well at eliminating the screen quality bias (revised Fig. S5c). Importantly though, this post-hoc correction again did not improve other downstream benchmark analyses, but rather made them somewhat worse compared with using the original D1 gene dependencies (revised Fig. S5d). We believe this can be explained both by the other improvements of the D2 model over D1, as well as the advantage provided by estimating these normalization terms as part of the D2 model. For example, D2 is able to utilize the screen-quality differences across cell lines to improve its estimates of gene dependencies by using the particular pattern of shRNA depletion across cell lines associated with their variable screen signal to help distinguish essential genes from off-target effects or other sources of noise.

The authors describe in detail how the binary shRNA-to-gene mapping G is computed. They should also describe how to compute the shRNA-to-seed mapping B . The authors do not mention that and the method used for G does not seem to be transferable to B .

As we now explain more clearly in the methods (section "Model description", pg. 27), the elements B_{is} (where i indexes shRNAs and s indexes unique 7mer seed sequences) are equal to one if sequence s is present in either of two seed regions in shRNA i , and 0 otherwise.

Figure S8c is missing a color key for the colors depicting the density of points.

We have added the color legend to this figure.

On page 25 the sentence fragment "the D1 model in several ways, as described below (and illustrated schematically in Fig." was accidentally copy and pasted in the next sentence and appears twice.

We have corrected this error, and thank the reviewer for catching this.

In general, the paper is well written, sound, and relevant to the perturbation community. However, we cannot recommend the paper for publication before the authors address the above concerns, especially on simulations and comparisons to other approaches.

Reviewer #2 (Remarks to the Author):

McFarland et al describe a new algorithm, DEMETER2, for the analysis of RNAi (and CRISPR) screens. Significant resources have been invested in the shRNA-based pooled library screening of cancer cell lines to identify "synthetic lethal" targets for therapeutic intervention. Even with the substantial advantage that CRISPR screening brings to bear on this problem, efforts to improve the analysis of the very large number of shRNA screens available are well justified.

The authors present a comprehensive upgrade to DEMETER that includes parameters for overall screen quality (itself estimated by evaluating the screen's ability to separate reference sets of essential and nonessential/nonexpressed genes), variable gene-level and seed-level (i.e. off-target) effects, and provides an estimate of absolute fitness defect for each gene. All of these are critical improvements that contribute to the impact of this algorithm.

Overall this study represents a important contribution to the field. It identifies concepts that are key weaknesses in the existing literature/methods, addresses them in a statistically robust manner, and potentially rescues a lot of data from oblivion.

Minor edit:

Fig 2b: the use of Spearman correlations may render this point moot, but the robustness of the Achilles correlation should be evaluated with regard to the removal of the two outliers (screens w/AGO2 expression < 0).

We have verified that the Spearman correlation for the Achilles data shown in Fig. 2b is virtually unchanged after removal of the two outlier points. While we certainly agree with the reviewer that this is an important point, we think the use of Spearman correlations in this instance should be sufficient to control for misleading results due to these outliers. We would be happy to modify the text to make this point more explicit if the reviewer believes this would be helpful.

Reviewer #3 (Remarks to the Author):

Summary: The authors have developed Demeter2, a significantly improved implementation of their previous method Demeter, to correct for technical and batch artifacts in RNAi screens of cancer cell lines. Despite some inherent limitations, cell line-based RNAi screens provide an extremely valuable resource to the community for understanding genetic interactions and their relevance to cancer. Thus strategies to aggregate and denoise these data are of great interest and value to the broader biomedical research community. Reducing signal due to off target effects and other technical noise that contaminates such datasets is a challenging problem. The authors convincingly demonstrate that Demeter2 improves the quality of RNAi screen data beyond what was possible with other existing methods. While this work builds on a previously published tool, the modifications result in a large overall improvement and allow novel quantification of gene dependencies relative to the previous incarnation. Overall the manuscript is clearly written and represents an important advance, and importantly both the source code and data will be made available to the research community. This reviewer has only a few suggestions.

Major comments:

It is exciting to see that Demeter2 improves over other available methods. After convincingly establishing the improvement in performance, the authors spend some time showing how different genomic features and mechanisms correspond to gene dependencies. The majority of improvement seems to come with more confident identification of CYCLOPs effects. The opportunity exists here to go into more depth in characterizing the gene dependencies that are recovered by the Demeter2 approach. Is it a set of genes that confer a smaller effect size of dependency, or that the same genes being detected in new cell lines? Are these genes mostly known essential genes, or are novel genes implicated? Is there any functional coherence across these smaller effect-size dependencies? While this manuscript is clearly more technical in spirit, any biological insights that might come from the ability to detect smaller effect sizes could broaden the interest in the manuscript.

We agree with the reviewer that further exploration of the biological results revealed by DEMETER2 would broaden interest in the manuscript. In the revised manuscript we have thus added several additional analyses to Figure 5 to better illustrate the nature of the biological relationships revealed by DEMETER2. In particular, in Fig. 5f we show a systematic analysis of correlations between gene dependency and gene 'dosage' (based on copy number and mRNA levels) as assessed by D2 in the combined data versus employing D1 on the Achilles or DRIVE datasets. This analysis reveals that the combined D2 dataset provides consistent moderate increases in dose-dependency correlations across a broad range of putative CYCLOPS genes, and in some cases (e.g. RPS29) the improvement provided by D2 is quite dramatic. Additionally, in the revised Fig. 5g we show that most of the feature-dependency relationships uniquely identified by the combined D2 dataset (that were not present in either D1 dataset) are either for common-essential genes (i.e. a majority of the CYCLOPS relationships), or they are relatively weaker associations that are revealed by the increased power of the combined dataset (most of the newly identified 'paralog loss' dependencies). We believe these analyses illustrate the

dominant factors (improved estimation of dependency profiles for essential genes, and increased power to resolve weaker effects) driving the improvements shown in Fig. 5d,e with the combined D2 dataset.

Minor comments:

Given that using D2 to combine datasets allowed assessment of the improvement in performance relative to the amount of data gained, it would be interesting to investigate the point where performance due to added data saturates for genes with different magnitudes of dependency. We believe that this point may be largely addressed by our response to reviewer 1, where we have added extensive analysis of the DEMETER2 model performance in simulations. Particularly note the newly added **Figs. S12f-h**, where we show improvement in model performance as a function of the number of cell lines included in the simulated dataset. While applying a similar type of analysis to the real data could be of interest as well, we believe there are several factors that complicate such an effort. Firstly, the gains provided by adding additional cell lines will be largely dependent on the details of the datasets being combined (e.g., Are the screens from non-overlapping sets of cell lines? Do they use the same or different libraries? Which types of cell lines compose each part of the dataset, etc.). Additionally, the results of such analyses will be highly dependent on the performance metrics, and statistical criteria, one applies (e.g. power to detect rare dependencies is expected to increase with very large sample sizes, but confident detection of dependency-biomarker relationships may require many fewer cell lines). Hence, we focus on the more straightforward technical assessment of model performance vs. dataset size using simulations.

Figure S8d was confusing at first since it wasn't clear that D2 figures previously were always accounting for uncertainty. It might be more clear if the description were framed as D2's performance being modestly but significantly reduced when precision is not used to weight dependency estimates.

We thank the reviewer for pointing out this source of confusion, and we have revised the description in the figure legend (and our reference to it in the main text) accordingly.

Figure 5b – add variance explained to the axis by each PC, or in the caption, for the different PCA plots

We have added the % variance explained to the axis labels in the revised **Fig. 5b**, as suggested.

Since the resource will be made publicly available, it would be helpful to include a level description of its composition, for example, an overview of the cell type composition of the cell lines.

We agree that this is an important piece of information to highlight, and have thus added a pie chart showing the composition of the combined DEMETER2 dataset by primary disease type in the revised **Fig. 5a**.

Additional corrections/changes:

In addition to addressing the reviewers' comments as described above, we noticed a few errors and oversights in our original manuscript during the revision process, which we have corrected in the revised version, as described below.

1. In the original dataset there were two cell lines (SUM52_BREAST and SUM52PE_BREAST), screened in the Marcotte and DRIVE datasets respectively, that we originally treated as separate cell lines. During the revision process we discovered

- that they are actually synonyms for the same cell line. Hence, we have rerun our analyses treating these as a single cell line that was screened in both datasets. All analyses, text, and publically available datasets have been updated accordingly. Note that this changes the total number of unique cell lines screened in the combined DEMETER2 dataset from 713, as we originally reported, to 712.
2. We removed a small number of non-human genes (e.g. GFP, RFP) from the shRNA-to-gene mapping file used by our models. This constituted a small portion (0.07%) of total mappings, and had a negligible influence on our results. This correction was included in the updated version of the publically available data.
 3. In the section “Integration of multiple RNAi datasets”, we had mistakenly reported that the combined DEMETER2 dataset also provides substantial increases in the number of identified ‘interacting protein’ relationships compared to the DEMETER1 datasets. This text should have referred to ‘paralog loss’ relationships, rather than ‘interacting protein’ relationships. Additionally, we realized that some ‘paralog loss’ relationships were mistakenly being counted also as ‘interacting protein’ relationships, which we have fixed in the revision.
 4. While we originally reported that combining separate DEMETER1 results across datasets suffered from ‘batch effect’ problems (similarly to pooling data and computing per-gene averages), we since realized that normalizing each DEMETER1 dataset before averaging them together largely mitigates any batch-related differences. Further, our analysis of the top principal components in the DEMETER1 data (in response to reviewer 1’s suggestion), showed that the high variance accounted for by the top principal component is largely due to differences in screen-quality across cell lines. In the revised manuscript we have thus clarified this point, showing that the large batch effects present when combining datasets are predominantly isolated to methods (other than DEMETER2) that estimate ‘absolute gene dependency’ (see revised text on pgs 16-17, and the revised **Fig. S9**). Nevertheless, despite the fact that relative dependency scores (such as DEMETER1) can be combined across datasets without the same batch-effect problems, the advantages of DEMETER2 compared with DEMETER1 that we show throughout the manuscript (e.g. absolute vs relative dependency scores, removal screen-quality related biases, and improvement of downstream benchmark analyses) still apply when combining datasets. This point is shown in the revised **Fig. S9**.

Reviewers' Comments:

Reviewer #1:

Remarks to the Author:

McFarland et al. have sufficiently addressed the comments on identifiability and overfitting. They have provided a good explanation on the choice of hyperparameters and the need for constraints on parameters for identifiability. Further, they have also shown robust performance on simulated data.

Additionally, the authors were asked to compare to other existing methods, which was addressed by comparing to MAGeCK. In the rebuttal, the authors argue that "It's unlikely that ScreenBEAM (or other previously described methods) will address the main challenges described here with large-scale RNAi screens, since it does not model RNAi off-target effects specifically, and treats data from each cell line independently." This might not be necessarily true if one were to use these methods in conjunction with off-target correcting methods, which was the original suggestion. Instead the authors only address "post-hoc" correction for screen quality bias by removing the first principal component in D1 and argue that this method fails in the downstream analysis. While the strength of D2 lies in the fact that it accounts for multiple sources of errors and biases in a single model and combines different data sources efficiently, one could always use a combination of existing tools to address the same goals/issues. Therefore, without a concrete comparative analysis, asserting that other ensemble methods would fail would be unfair.

Does the removal of the first PC of the gene dependency matrix actually relate to the multi-stage model? It does not seem apparent, that this is equivalent to using a method to estimate off-target effects and another one to subsequently estimate gene dependencies. Also, are the cell line average dependency scores for D1-PC the averages over components 2 to n? In Fig.S5c D1-PC seems to perform better than D2 and even though the test statistics suggests otherwise Fig. S5d seems not to convincingly argue against D1-PC either. Especially for the DRIVE data set, D1-PC seems to perform significantly better than D1 (the red curve is below the green curve almost everywhere except for a small interval). It almost seems as if a two tailed test statistic was used instead of testing, if one was larger than the other.

Minor:

The color key of figure S8c does not indicate any values.

Reviewer #2:

Remarks to the Author:

The authors have addressed my concerns, and especially those of the other reviewers, in substantial depth.

Reviewer #3:

Remarks to the Author:

This reviewer has no further comments.

Reviewer #1 (Remarks to the Author):

McFarland et al. have sufficiently addressed the comments on identifiability and overfitting. They have provided a good explanation on the choice of hyperparameters and the need for constraints on parameters for identifiability. Further, they have also shown robust performance on simulated data.

Additionally, the authors were asked to compare to other existing methods, which was addressed by comparing to MAGeCK. In the rebuttal, the authors argue that “It’s unlikely that ScreenBEAM (or other previously described methods) will address the main challenges described here with large-scale RNAi screens, since it does not model RNAi off-target effects specifically, and treats data from each cell line independently.” This might not be necessarily true if one were to use these methods in conjunction with off-target correcting methods, which was the original suggestion. Instead the authors only address “post-hoc” correction for screen quality bias by removing the first principal component in D1 and argue that this method fails in the downstream analysis. While the strength of D2 lies in the fact that it accounts for multiple sources of errors and biases in a single model and combines different data sources efficiently, one could always use a combination of existing tools to address the same goals/issues. Therefore, without a concrete comparative analysis, asserting that other ensemble methods would fail would be unfair.

We agree with the reviewer that we cannot rule out the possibility that alternative approaches that combine multiple existing methods might be used to address the main challenges we describe. We did not intend to assert otherwise, and could not find anywhere in the manuscript text where we did so. Indeed, we designed the D2 model itself by starting from the previous D1 model and systematically adding additional features to address these specific challenges. Although it might seem simpler to consider ‘ensemble’ approaches that separately apply a series of corrections to address each specific challenge in turn, we believe there are significant advantages to utilizing a unified statistical model to address these issues. We are also not aware of how one would actually combine separate existing methods to address all of these problems. For example, in the manuscript we consider a pipeline where D1 is first applied to address off-target effects and variable shRNA efficacy, and then PCA analysis is separately applied to the D1 scores to identify and remove biases related to variable screen quality. We found that such a hybrid approach actually had a negative impact on downstream analyses (compared with the original D1 scores), and one would still need to introduce elements into such an ensemble method allowing it to estimate dependencies on an absolute scale, integrate multiple screening datasets in a statistically principled way, and ideally provide uncertainty estimates for gene dependencies, all of which are accomplished by D2. To help clarify this point, we have added several sentences to the revised Discussion section (pg. 18).

Does the removal of the first PC of the gene dependency matrix actually relate to the multi-stage model? It does not seem apparent, that this is equivalent to using a method to estimate off-target effects and another one to subsequently estimate gene dependencies.

We show (Fig. S5) that the first PC of the D1-estimated gene dependency matrix accurately captures the variation in screen signal across cell lines (in both the Achilles and DRIVE datasets). Thus, combining D1 for estimating gene dependencies with removal of this PC component to correct for screen quality bias represents a potential ‘multi-stage’ modeling strategy.

Also, are the cell line average dependency scores for D1-PC the averages over components 2 to n?

We do not, in fact, compute the across-cell-line average dependency scores for the D1-PC method (D1 does not estimate the across-cell-line average dependency score, but rather sets it to 0 explicitly). The per-cell line dependency scores for the D1-PC method are computed not by averaging over components 2 to n, but rather by linearly projecting out the first PC from the matrix of dependency scores:

$\mathbf{X}_{(-1)} = \mathbf{X} - \mathbf{X}\mathbf{w}_{(1)}\mathbf{w}_{(1)}^T$, where \mathbf{X} is the matrix of dependency scores, $\mathbf{w}_{(1)}$ is its first principal component, and $\mathbf{X}_{(-1)}$ represents the matrix after projecting out the first PC.

In Fig.S5c D1-PC seems to perform better than D2 and even though the test statistics suggests otherwise Fig. S5d seems not to convincingly argue against D1-PC either. Especially for the DRIVE data set, D1-PC seems to perform significantly better than D1 (the red curve is below the green curve almost everywhere

except for a small interval). It almost seems as if a two tailed test statistic was used instead of testing, if one was larger than the other.

We agree with the reviewer that Fig. S5c suggests that the D1-PC method more completely removes any relationship between gene dependencies and *AGO2* expression compared with D2. We believe that this is due to D2 using a more 'constrained' model for removing this screen-quality bias compared with the D1-PC approach. In essence, the D1-PC model is able to estimate a separate term for each gene specifying how much its dependency profile is affected by the variation in screen quality across cell lines. In contrast, D2 models the effect of variable screen signal on each gene's dependency profile as a multiplicative interaction (such that the degree of screen quality bias reflected in a gene's dependency profile is determined by how essential the gene is on average). This assumption may fail to capture some more complex relationships or gene-specific behaviors, but it also helps the D2 model estimate absolute dependency, and we found it provides improved results in downstream analyses. Nevertheless, in the revised manuscript (Fig. S5 caption) we have explicitly mentioned this difference, and the potential explanation for it.

Regarding Fig. S5d, while we agree with the reviewer that the difference appears visually small (particularly for the DRIVE dataset), we have verified that our statistical analyses were correct. For the DRIVE data, while the mean correlation magnitude is not significantly different between D1-PC and D1 ($p = 0.51$, paired t-test, $n = 231$ feature-dependency pairs), the median is significantly smaller for D1-PC ($p = 1.6E-3$, Wilcoxon signed rank test). Thus, the difference is due to a small but consistent shift in the mode of the distribution, which is indeed difficult to resolve in the previous figure. We have expanded the limits of the y-axes in this figure to better visualize the differences in these distributions.

Minor:

The color key of figure S8c does not indicate any values.

We had initially left out values for the color key in this figure because the units of the density estimates are difficult to interpret, and we were worried they might confuse readers. We agree with the reviewer, however, that this could be problematic, and thus have introduced labels to indicate 'low' and 'high' density on the color spectrum (in Fig. S8c, as well as other figures where we use a color map to represent the data point density: e.g. Fig. 3a-b).

Reviewer #2 (Remarks to the Author):

The authors have addressed my concerns, and especially those of the other reviewers, in substantial depth.

Reviewer #3 (Remarks to the Author):

This reviewer has no further comments.